# CONTINUOUS-TIME FLOWS FOR EFFICIENT INFERENCE AND DENSITY ESTIMATION

## ABSTRACT

Two fundamental problems in unsupervised learning are efficient inference for latent-variable models and robust density estimation based on large amounts of unlabeled data. For efficient inference, normalizing flows have been recently developed to approximate a target distribution arbitrarily well. In practice, however, normalizing flows only consist of a finite number of deterministic transformations, and thus they possess no guarantee on the approximation accuracy. For density estimation, the generative adversarial network (GAN) has been advanced as an appealing model, due to its often excellent performance in generating samples. In this paper, we propose the concept of *continuous-time flows* (CTFs), a family of diffusion-based methods that are able to asymptotically approach a target distribution. Distinct from normalizing flows and GANs, CTFs can be adopted to achieve the above two goals in one framework, with theoretical guarantees. Our framework includes distilling knowledge from a CTF for efficient inference, and learning an explicit energy-based distribution with CTFs for density estimation. Experiments on various tasks demonstrate promising performance of the proposed CTF framework, compared to related techniques.

## 1 INTRODUCTION

Efficient inference and robust density estimation are two important goals in unsupervised learning. In fact, inference and density estimation can be unified from the perspective of learning desired target distributions. In inference problems, one is given an unnormalized distribution (*e.g.*, the posterior distribution), and the goal is to learn a distribution that is close to the unnormalized distribution. In density estimation, one tries to learn an unknown data distribution given samples from it. It is also helpful to make a distinction between two types of representations for learning distributions: explicit and implicit methods (Mohamed & Lakshminarayanan, 2017). Explicit methods provide a prescribed parametric form for the distribution, while implicit methods learn a stochastic procedure to directly generate samples from the unknown distribution.

Existing deep generative models can easily be identified from this taxonomy. For example, the standard variational autoencoder (VAE) (Kingma & Welling, 2014; Rezende et al., 2014) is an important example of an *explicit inference* method. Within the inference arm (encoder) of a VAE, recent research has focused on improving the accuracy of the approximation to the posterior distribution on latent variables (codes) using normalizing flow (Rezende & Mohamed, 2015). Normalizing flow is particularly interesting due to its ability to approximate the posterior distribution arbitrarily well, while maintaining explicit parametric forms. On the other hand, Stein VAE (Feng et al., 2017; Pu et al., 2017b) is an *implicit inference* method, as it only learns to draw samples to approximate posteriors, without assuming an explicit form for the distribution.. For density estimation on observed data, the generative adversarial network (GAN) can be regarded as an *implicit density estimation* method (Ranganath et al., 2016; Huszár, 2017; Mohamed & Lakshminarayanan, 2017), in the sense that one may sample from the distribution (regarded as a representation of the unknown distribution), but an explicit form for the distribution is not estimated. GAN has recently been augmented by Flow-GAN (Grover et al., 2017) to incorporate a likelihood term for *explicit density estimation*. Further, there also are some works trying to perform inference within the implicit density estimation framework, *e.g.*, the real-valued non-volume preserving (real NVP) transformations algorithm (Dinh et al., 2017) was proposed as a tractable yet expressive approach to model high-dimensional data.

Some aforementioned methods rely on the concept of *flows*. A flow defines a series of transformations for a random variable (RV), such that the distribution of the RV evolves from a simple distribution to a more complex distribution. When the sequence of transformations are indexed on a discrete-time domain (*e.g.*, indexed with integers) with a finite number of transformations, this method is referred to as a normalizing flow (Rezende & Mohamed, 2015). Various efficient implementations of normalizing flows have been proposed, such as the planar, radial (Rezende & Mohamed, 2015), Householder (Tomczak & Welling, 2016), and inverse autoregressive flows (Kingma et al., 2016). One theoretical limitation of existing normalizing flows is that there is no guarantee on the approximation accuracy due to the finite number of transformations.

By contrast, little work has explored the applicability of continuous-time flows (CTFs) in deep generative models, where a sequence of transformations are indexed on a continuous-time domain (*e.g.*, indexed with real numbers). There are at least two reasons encouraging research in this direction: *i*) CTFs are more general than traditional normalizing flows in terms of modeling flexibility, due to the intrinsic infinite number of transformations; *ii*) CTFs are more theoretically grounded, in the sense that they are guaranteed to approach a target distribution asymptotically (details provided in Section 2.2). Unfortunately, these advantages also bring challenges for efficient learning, in that: *i*) it is difficult to optimize over the variational lower bound in the inference framework, due to the extra randomness introduced in CTFs; *ii*) it is difficult to design algorithms for efficient learning of CTF-based models, due to the induced infinite number of transformations.

In this paper, we propose efficient ways to apply CTFs for the two motivating tasks. Based on the CTF, our framework learns to drawn samples directly from desired distributions (*e.g.*, the unknown posterior and data distributions) for both inference and density estimation tasks. In addition, we are able to learn an explicit form of the unknown data distribution for density estimation*. This shares a similar flavor as Wang & Liu (2017); Feng et al. (2017). Specifically, *i*) for efficient inference, we first show that optimizing the variational lower bound with CTFs can be achieved by decomposing the optimization problem into a sequence of sub-optimization problems, based on a variational formulation of the Fokker-Planck equations from statistical physics (Jordan et al., 1998). Based on this decomposition, we derive bounds on the approximation errors when applying numerical methods to solve a CTF. For computational efficiency, we generalize ideas from Gershman & Goodman (2014) to distill knowledge of a CTF into an efficient inference network; *ii*) for density estimation, we propose to use a flexible Gibbsian-style distribution (implemented by a deep neural network) to model an unknown data distribution, whose samples can be drawn by learning a stochastic generator with our CTF framework. The Gibbsian-style data distribution and the stochastic generator are learned alternatively, leading to a learning procedure that is connected to the GAN framework (Goodfellow et al., 2014), but that yields an explicit distribution for the data. We conduct a number of experiments on real datasets, demonstrating excellent performance of the proposed framework, relative to existing representative approaches.

## 2 PRELIMINARIES

We first review related techniques of performing efficient inference and density estimation in the machine learning literature. We then introduce the general concept of continuous-time flows.

### 2.1 EFFICIENT INFERENCE AND DENSITY ESTIMATION

**Efficient inference with normalizing flows** Consider a probabilistic generative model with observation $\mathbf{x} \in \mathbb{R}^D$ and latent variable $\mathbf{z} \in \mathbb{R}^L$ such that $\mathbf{x} \mid \mathbf{z} \sim p_{\boldsymbol{\theta}}(\mathbf{x} \mid \mathbf{z})$ with $\mathbf{z} \sim p(\mathbf{z})$. For efficient inference of $\mathbf{z}$, the VAE (Kingma & Welling, 2014) introduces the concept of an inference network (recognition model or encoder), $q_{\boldsymbol{\phi}}(\mathbf{z} \mid \mathbf{x})$, as a variational distribution in the VB framework. An inference network is typically a stochastic (nonlinear) mapping from the input $\mathbf{x}$ to the latent $\mathbf{z}$, with associated parameters $\boldsymbol{\phi}$. For example, one of the simplest inference networks is defined as $q_{\boldsymbol{\phi}}(\mathbf{z} \mid \mathbf{x}) = \mathcal{N}(\mathbf{z}; \boldsymbol{\mu}_{\boldsymbol{\phi}}(\mathbf{x}), \mathrm{diag}(\boldsymbol{\sigma}_{\boldsymbol{\phi}}^2(\mathbf{x})))$, where the mean function $\boldsymbol{\mu}_{\boldsymbol{\phi}}(\mathbf{x})$ and the standard-derivation function $\boldsymbol{\sigma}_{\boldsymbol{\phi}}(\mathbf{x})$ are specified via deep neural networks parameterized by $\boldsymbol{\phi}$. Parameters are learned by minimizing the evidence lower bound (ELBO), *i.e.*, the KL divergence between $p_{\boldsymbol{\theta}}(\mathbf{x}, \mathbf{z})$

---

*Although the density is represented as an energy-based distribution with an intractable normalizer.

and $q_{\phi}(\mathbf{z} \mid \mathbf{x})$: KL $(q_{\phi}(\mathbf{z} \mid \mathbf{x}) \| p_{\theta}(\mathbf{x}, \mathbf{z})) \triangleq \mathbb{E}_{q_{\phi}(\mathbf{z} \mid \mathbf{x})} [\log q_{\phi}(\mathbf{z} \mid \mathbf{x}) - \log p_{\theta}(\mathbf{x}, \mathbf{z})]$, via stochastic gradient descent (Bottou, 2012).

One limitation of the VAE framework is that $q_{\phi}(\mathbf{z} \mid \mathbf{x})$ is often restricted to simple distributions for feasibility, *e.g.*, the normal distribution discussed above, and thus the gap between $q_{\phi}(\mathbf{z} \mid \mathbf{x})$ and $p_{\theta}(\mathbf{z} \mid \mathbf{x})$ is typically large for complicated posterior distributions. Normalizing flows is a recently proposed VB-based technique designed to mitigate this problem (Rezende & Mohamed, 2015). The idea is to augment $\mathbf{z}$ via a sequence of deterministic invertible transformations $\{\mathcal{T}_k : \mathbb{R}^L \to \mathbb{R}^L\}_{k=1}^K$, such that: $\mathbf{z}_0 \sim q_{\phi}(\cdot \mid \mathbf{x}), \mathbf{z}_1 = \mathcal{T}_1(\mathbf{z}_0), \cdots, \mathbf{z}_K = \mathcal{T}_K(\mathbf{z}_{K-1})$.

Note the transformations $\{\mathcal{T}_k\}$ are typically endowed with different parameters, and we absorb them into $\phi$. Because the transformations are deterministic, the distribution of $\mathbf{z}_K$ can be written as $q(\mathbf{z}_K) = q_{\phi}(\mathbf{z}_0 \mid \mathbf{x}) \prod_{k=1}^K \left| \det \frac{\partial \mathcal{T}_k}{\partial \mathbf{z}_k} \right|^{-1}$ via the change of variable formula. As a result, the ELBO for normalizing flows becomes:

$$\text{KL} \left( q_{\phi}(\mathbf{z}_K \mid \mathbf{x}) \| p_{\theta}(\mathbf{x}, \mathbf{z}) \right) = \tag{1}$$

$$\mathbb{E}_{q_{\phi}(\mathbf{z}_0 \mid \mathbf{x})} [\log q_{\phi}(\mathbf{z}_0 \mid \mathbf{x})] - \mathbb{E}_{q_{\phi}(\mathbf{z}_0 \mid \mathbf{x})} [\log p_{\theta}(\mathbf{x}, \mathbf{z}_K)] - \mathbb{E}_{q_{\phi}(\mathbf{z}_0 \mid \mathbf{x})} \left[ \sum_{k=1}^K \log \left| \det \frac{\partial \mathcal{T}_k}{\partial \mathbf{z}_k} \right| \right] .$$

Typically, transformations $\mathcal{T}_k$ of a simple parametric form are employed to make the computations tractable (Rezende & Mohamed, 2015). Our method for inference generalizes these discrete-time transformation to continuous-time transformations, ensuring convergence of the transformations to the target distribution.

**Density estimation overview** There exist implicit and explicit density-estimation methods. Implicit density models such as GAN provide a flexible way to draw samples directly from unknown data distributions (via a deep neural network (DNN) called a generator with stochastic inputs) without explicitly modeling their density forms; whereas explicit models such as the pixel RNN/CNN (van den Oord et al., 2016) define and learn explicit forms of the unknown data distributions. This gives the advantage that the likelihood for a test data point can be explicitly evaluated. However, the generation of samples is typically time-consuming due to the sequential generation nature.

Similar to Wang & Liu (2017), our CTF-based approach in Section 4 provides an alternative way for this problem, by simultaneously learning an explicit Gibbsian-style data distribution (estimated density) and a generator whose generated samples match the learned Gibbsian distribution. This not only gives us the advantage of explicit density modeling but also provides an efficient way to generate samples.

## 2.2 CONTINUOUS-TIME FLOWS

We notice two potential limitations with traditional normalizing flows: *i*) given specified transformations $\{\mathcal{T}_k\}$, there is no guarantee that the distribution of $\mathbf{z}_K$ could exactly match $p_{\theta}(\mathbf{x}, \mathbf{z})$; *ii*) the randomness is only introduced in $\mathbf{z}_0$ (from the inference network), limiting the representation power. We specify CTFs where the transformations are indexed by real numbers, thus they could be considered as consisting of an infinite number of transformations. Further, we consider stochastic flows where randomness is injected in a continuous-time manner. In fact, the concept of CTFs (such as the Hamiltonian flow) has been introduced in Rezende & Mohamed (2015), without further development on efficient inference.

We consider a flow on $\mathbb{R}^L$, defined as the mapping[†] $\mathcal{T} : \mathbb{R}^L \times \mathbb{R} \to \mathbb{R}^L$ such that[‡] we have $\mathcal{T}(\mathbf{Z}, 0) = \mathbf{z}$ and $\mathcal{T}(\mathcal{T}(\mathbf{Z}, t), s) = \mathcal{T}(\mathbf{Z}, s + t)$, for all $\mathbf{Z} \in \mathbb{R}^L$ and $s, t \in \mathbb{R}$. A typical example of this family is defined as $\mathcal{T}(\mathbf{Z}, t) = \mathbf{Z}_t$, where $\mathbf{Z}_t$ is driven by a diffusion of the form:

$$d\mathbf{Z}_t = F(\mathbf{Z}_t)dt + V(\mathbf{Z}_t)d\mathcal{W} . \tag{2}$$

---

[†] We reuse the notation $\mathcal{T}$ as transformations from the discrete case above for simplicity, and use $\mathbf{Z}$ instead of $\mathbf{z}$ (reserved for the discrete-time setting) to denote the random variable in the continuous-time setting.

[‡] Note we define continuous-time flows in terms of latent variable $\mathbf{Z}$ in order to incorporate it into the setting of inference. However, the same description applies when we define the flow in data space, which is the setting of density estimation in Section 4.

Here $F : \mathbb{R}^L \rightarrow \mathbb{R}^L$, $V : \mathbb{R}^{L \times L} \rightarrow \mathbb{R}^L$ are called the drift term and diffusion term, respectively; $\mathcal{W}$ is the standard $L$-dimensional Brownian motion. In the context of inference, we seek to make the stationary distribution of $\mathbf{Z}_t$ approach $p_{\boldsymbol{\theta}}(\mathbf{z} \,|\, \mathbf{x})$. One solution for this is to set $F(\mathbf{Z}_t) = \frac{1}{2} \nabla_{\mathbf{z}} \log p_{\boldsymbol{\theta}}(\mathbf{x}, \mathbf{z} = \mathbf{Z}_t)$ and $V(\mathbf{Z}_t) = \mathbf{I}_L$ with $\mathbf{I}_L$ the $L \times L$ identity matrix. The resulting diffusion is called Langevin dynamics Welling & Teh (2011). Denoting the distribution of $\mathbf{Z}_t$ as $\rho_t$, it is well known Risken (1989) that $\rho_t$ is characterized by the Fokker-Planck (FP) equation:

$$\frac{\partial \rho_t}{\partial t} = -\nabla_{\mathbf{z}} \cdot (\rho_t F(\mathbf{Z}_t)) + \nabla_{\mathbf{z}} \nabla_{\mathbf{z}} : \left( \rho_t V(\mathbf{Z}_t) V^\top (\mathbf{Z}_t) \right) \ , \tag{3}$$

where $\mathbf{a} \cdot \mathbf{b} \triangleq \mathbf{a}^\top \mathbf{b}$ for vectors $\mathbf{a}$ and $\mathbf{b}$, $\mathbf{A} : \mathbf{B} \triangleq \text{trace}(\mathbf{A}^\top \mathbf{B})$ for matrices $\mathbf{A}$ and $\mathbf{B}$.

For simplicity, we consider the flow defined by the Langevin dynamics specified above, though our results generalize to other stochastic flows Dorogovtsev & Nishchenko (2014). In the following, we specify the ELBO under a CTF, which can then be readily solved by a discretized numerical scheme, based on the results from Jordan et al. (1998). An approximation error bound for the scheme is also derived. We defer proofs of our theoretical results to the Supplementary Material (SM) for conciseness.

# 3 CONTINUOUS-TIME FLOWS FOR INFERENCE

We first give an overview of our CTF-based method for efficient inference. We adopt the VAE/normalizing-flow framework with an encoder-decoder structure. An important difference is that instead of feeding data to an encoder and sampling a latent representation in the output as in VAE, we concatenate the data with independent noise as input and directly generate output samples[§]. These output samples are then driven by the CTF to approach the true posterior distribution. In the learning process, the implicit transformations from the CTF are sequentially distilled into the inference network by amortized learning, making the inference network flexible enough to represent the true posterior distribution. In the following subsections, we specify our framework in detail.

## 3.1 THE VARIATIONAL LOWER BOUND AND DISCRETIZED APPROXIMATION

We first incorporate CTF into the normalizing-flow framework by writing out the corresponding ELBO. Note that there are two steps in the inference process. First, an initial $\mathbf{z}_0$ is drawn from the inference network $q_{\boldsymbol{\phi}}(\cdot \,|\, \mathbf{x})$; second, $\mathbf{z}_0$ is evolved via a diffusion such as (2) for time $T$ (via the transformation $\mathbf{Z}_T = \mathcal{T}(\mathbf{z}_0, T)$). Consequently, the ELBO for CTF can be written as

$$\mathcal{F}(\mathbf{x}) = \mathbb{E}_{q_{\boldsymbol{\phi}}(\mathbf{z}_0 \,|\, \mathbf{x})} \mathbb{E}_{\rho_T} \left[ \log \rho_T - \log p_{\boldsymbol{\theta}}(\mathbf{x}, \mathbf{Z}_T) + \log \left| \det \frac{\partial \mathbf{Z}_T}{\partial \mathbf{z}_0} \right| \right] \triangleq \mathbb{E}_{q_{\boldsymbol{\phi}}(\mathbf{z}_0 \,|\, \mathbf{x})} \left[ \mathcal{F}_1(\mathbf{x}, \mathbf{z}_0) \right] \ . \tag{4}$$

Note the term $\mathcal{F}_1(\mathbf{x}, \mathbf{z}_0)$ is intractable to calculate, in that $i)$ $\rho_T$ does not have an explicit form; $ii)$ the Jacobian $\frac{\partial \mathbf{Z}_T}{\partial \mathbf{z}_0}$ is generally infeasible. In the following, we propose an approximate solution for problem $i)$. Learning by avoiding problem $ii)$ is presented in Section 3.2 via amortization.

For problem $i)$, a reformulation of the results from Jordan et al. (1998) leads to a nice way to approximate $\rho_t$ in Lemma 1. Note in practice we adopt an *implicit* method which uses samples to approximate the solution in Lemma 1 for feasibility, detailed in (6).

**Lemma 1.** *Assume that* $\log p_{\boldsymbol{\theta}}(\mathbf{x}, \mathbf{z}) \leq C_1$ *is infinitely differentiable, and* $\|\nabla_{\mathbf{z}} \log p_{\boldsymbol{\theta}}(\mathbf{x}, \mathbf{z})\| \leq C_2 (1 + C_1 - \log p_{\boldsymbol{\theta}}(\mathbf{x}, \mathbf{z})) \ (\forall \mathbf{x}, \mathbf{z})$ *for some constants* $\{C_1, C_2\}$. *Let* $T = hK$ ($h$ *is the stepsize in discretization and* $K$ *is the number of transformations*), $\rho_0 \triangleq q_{\boldsymbol{\phi}}(\mathbf{z}_0 \,|\, \mathbf{x})$, *and* $\{\tilde{\rho}_k\}_{k=1}^K$ *be the solution of the functional optimization problem:*

$$\tilde{\rho}_k = \arg \min_{\rho \in \mathcal{K}} KL \left( \rho \| p_{\boldsymbol{\theta}}(\mathbf{x}, \mathbf{z}) \right) + \frac{1}{2h} W_2^2 \left( \tilde{\rho}_{k-1}, \rho \right) \ , \tag{5}$$

*where* $W_2^2 (\mu_1, \mu_2) \triangleq \inf_{p \in \mathcal{P}(\mu_1, \mu_2)} \int \|\mathbf{x} - \mathbf{y}\|_2^2 \, p(\mathrm{d}\,\mathbf{x}, \mathrm{d}\,\mathbf{y})$, $W_2 (\mu_1, \mu_2)$ *is the 2nd-order Wasserstein distance, with* $\mathcal{P}(\mu_1, \mu_2)$ *being the space of joint distributions on* $\{\mu_1, \mu_2\}$. $\mathcal{K}$ *is the space of probability distributions with the finite 2nd-order moment. Then* $\tilde{\rho}_K$ *converges to* $\rho_T$ *in the limit of* $h \rightarrow 0$, *i.e.,* $\lim_{h \rightarrow 0} \tilde{\rho}_K = \rho_T$, *where* $\rho_T$ *is the solution of the FP equation* (3) *at time* $T$.

---

[§]Such structure can represent much more complex distributions than a parametric form, useful for following procedures. We argue an explicit distribution form is not as important in inference as that in density estimation.

Lemma 1 reveals an interesting way to compute $\rho_T$ via a sequence of functional optimization problems. By comparing it with the objective of the traditional normalizing flow, which minimizes the KL-divergence between $\rho_K$ and $p_{\boldsymbol{\theta}}(\mathbf{x}, \mathbf{z})$, at each sub-optimization-problem in Lemma 1, it minimizes the KL-divergence between $\tilde{\rho}_k$ and $p_{\boldsymbol{\theta}}(\mathbf{x}, \mathbf{z})$, plus a regularization term as the Wasserstein distance between $\tilde{\rho}_{k-1}$ and $\tilde{\rho}_k$. The extra Wasserstein-distance term arises naturally due to the fact that the Langevin diffusion can be explained as a gradient flow whose geometry is equipped with the Wasserstein distance (Otto, 1998). From another point of view, it is known that the Wasserstein distance is a better metric for probability distributions than the KL-divergence, especially in the case of non-overlapping domains (Arjovsky & Bottou, 2017; Arjovsky et al., 2017). By using the Wasserstein term as a regularizer, the CTF alleviates the issue in non-overlapping domains by introducing the Brownian-motion (noise) term in the evolution (2). This relates to the idea in (Arjovsky & Bottou, 2017), in which noise is added in parameter updates to alleviate the intrinsic drawback of the KL-divergence metric.

The optimization problem in Lemma 1 is difficult to deal with directly. In practice, we instead approximate the discretization in an equivalent way by simulation from the CTF. Starting from $\mathbf{z}_0$, $\mathbf{z}_k$ ($k = 0, \cdots, K - 1$) is fed into a transformation $\mathcal{T}_k$ (specified below), resulting in $\mathbf{z}_{k+1}$ whose distribution coincides with $\tilde{\rho}_{k+1}$ in Lemma 1. The discretization procedure is illustrated in Figure 1. We must specify the transformations $\mathcal{T}_k$. For each $k$, let $t = hk$; we can conclude from Lemma 1 that $\lim_{h \to 0} \tilde{\rho}_k = \rho_t$. From FP theory, $\rho_t$ is obtained by solving the diffusion (2) with initial condition $\mathbf{Z}_0 = \mathbf{z}_0$. It is thus reasonable to specify the transformation $\mathcal{T}_k$ as the $k$-th step of a numerical integrator for (2). Specifically, we specify $\mathcal{T}_k$ as a stochastic transformation:

$$\mathbf{z}_k = \mathcal{T}_k(\mathbf{z}_{k-1}) \triangleq \mathbf{z}_{k-1} + F(\mathbf{z}_{k-1})h + V(\mathbf{z}_{k-1})\boldsymbol{\zeta}_k , \tag{6}$$

where $\boldsymbol{\zeta}_k \sim \mathcal{N}(\mathbf{0}, h\mathbf{I}_L)$ is drawn from an isotropic normal. Note the transformation defined here is stochastic, thus we only get samples from $\tilde{\rho}_K$ at the end. A natural way to approximate $\tilde{\rho}_K$ is to use the empirical sample distribution, *i.e.*, $\tilde{\rho}_K \approx \frac{1}{K} \sum_{k=1}^{K} \delta_{\mathbf{z}_k} \triangleq \bar{\rho}_T$ with $\delta_{\mathbf{z}}$ a point mass at $\mathbf{z}$. Afterwards, $\tilde{\rho}_K$ (thus $\bar{\rho}_T$) will be used to approximate the true $\rho_T$ from (3).

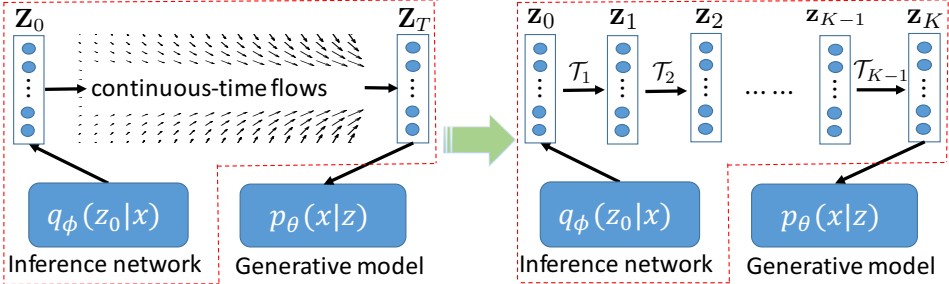

Figure 1: Discretized approximation (right) of a continuous-time flow (left). Densities $\{\tilde{\rho}_k\}$ of $\{\mathbf{z}_k\}$ evolve via transformations $\{\mathcal{T}_k\}$, with $\tilde{\rho}_k \to \rho_{hk}$ when $h \to 0$ for each $k$ due to Lemma 1.

Note that we use the simple sample averaging for the convenience of analysis, and the approximation for $\rho_T$ is not necessarily optimal. Better approximation can be obtained by assigning more weights to the more recent samples. However, this leads to more challenges in theoretical analysis, an interesting future direction to pursue. In the following, we study how well $\bar{\rho}_T$ approximates $\rho_T$. Following literature on numerical approximation for Itô diffusions (Vollmer et al., 2016; Chen et al., 2015), we consider a 1-Lipschitz test function $\psi : \mathbb{R}^L \to \mathbb{R}$, and use the mean square error (MSE) bound to measure the closeness of $\bar{\rho}_T$ and $\rho_T$, defined as: $\text{MSE}(\bar{\rho}_T, \rho_T; \psi) \triangleq \mathbb{E}\left(\int \psi(\mathbf{z})(\tilde{\rho}_T - \rho_T)(\mathbf{z})\mathrm{d}\,\mathbf{z}\right)^2$, where the expectation is taken over all the randomness in the construction of $\tilde{\rho}_T$. Note that our goal is related but different from the standard setup as in Vollmer et al. (2016); Chen et al. (2015), which studies the closeness of $\bar{\rho}_T$ to $p_{\boldsymbol{\theta}}(\mathbf{x}, \mathbf{z})$. We need to adopt the assumptions from Vollmer et al. (2016); Chen et al. (2015), which are described in the Supplementary Material (SM). The assumptions are somewhat involved but essentially require coefficients of the diffusion (2) to be well-behaved. We derive the following bound for the MSE of the sampled approximation, $\bar{\rho}_T$, and the true distribution.

**Theorem 2.** *Under Assumption 1 in the SM, assume that $\int \rho_T(\mathbf{z}) p_{\boldsymbol{\theta}}^{-1}(\mathbf{x}, \mathbf{z}) \mathrm{d}\,\mathbf{z} < \infty$ and there exists a constant $C$ such that $\frac{\mathrm{d}W_2^2(\rho_T, p_{\boldsymbol{\theta}}(\mathbf{x},\mathbf{z}))}{\mathrm{d}t} \geq C W_2^2\left(\rho_T, p_{\boldsymbol{\theta}}(\mathbf{x}, \mathbf{z})\right)$, the MSE is bounded as*

$$MSE(\bar{\rho}_T, \rho_T; \psi) = O\left(\frac{1}{hK} + h^2 + e^{-2ChK}\right) .$$

The last assumption in Theorem 2 requires $\rho_T$ to evolve fast through the FP equation, which is a standard assumption used to establish convergence to equilibrium for FP equations (Bolley et al., 2012). The MSE bound consists of three terms, the first two terms come from numerical approximation of the continuous-time diffusion, whereas the third term comes from the convergence bound of the FP equation in terms of the Wasserstein distance (Bolley et al., 2012). When the time $T = hK$ is large enough, the third term may be ignored due to its exponential-decay rate. Moreover, in the infinite-time limit, the bound endows a bias proportional to $h$; this, however, can be removed by adopting a decreasing-step-size scheme in the numerical method, as in standard stochastic gradient MCMC methods (Teh et al., 2016; Chen et al., 2015).

**Remark 3.** *To examine the optimal bound in Theorem 2, we drop out the term $e^{-2ChK}$ in the long-time case (when $hK$ is large enough) for simplicity because it is in a much lower order term than the other terms. The optimal MSE bound (over $h$) decreases at a rate of $O\left(K^{-2/3}\right)$, meaning that $O\left(\epsilon^{-3/2}\right)$ steps of transformations in Figure 1 (right) are needed to reach an $\epsilon$-accurate approximation, i.e., $MSE \leq \epsilon$. This is computationally expensive. An efficient way for inference is thus imperative, developed in the next section.*

### 3.2 Efficient inference via amortization

Even though we approximate $\rho_T$ with $\bar{\rho}_T$, it is still infeasible to directly apply it to the ELBO in (4) as $\bar{\rho}_T$ is discrete. To deal with this problem, we adopt the idea of "amortized learning" (Gershman & Goodman, 2014) for efficient inference. The main idea is to optimize the two sets of parameters $\boldsymbol{\phi}$ and $\boldsymbol{\theta}$ alternatively, based on different but related objective functions.

**Updating $\boldsymbol{\phi}$** To explain the idea, first note that the ELBO can be equivalently written as

$$\mathcal{F}(\mathbf{x}) = \mathbb{E}_{\rho_0 \triangleq q_{\boldsymbol{\phi}}(\mathbf{z}_0 \mid \mathbf{x})} \mathbb{E}_{\rho_T} \left[\log \rho_0 - \log p_{\boldsymbol{\theta}}(\mathbf{x}, \mathbf{Z}_T)\right] . \tag{7}$$

When $\rho_0 = \rho_T$, it is easy to see that: $\mathcal{F}(\mathbf{x}) = \mathbb{E}_{\rho_0}\left[\log \rho_0 - \log p_{\boldsymbol{\theta}}(\mathbf{Z}_T \mid \mathbf{x})\right] + \log p(\mathbf{x}) = \log p(\mathbf{x})$, which essentially makes the gap between $q_{\boldsymbol{\phi}}(\mathbf{z}_0 \mid \mathbf{x})$ and $p_{\boldsymbol{\theta}}(\mathbf{Z}_T \mid \mathbf{x})$ vanished. As a result, our goal is to learn $\boldsymbol{\phi}$ such that $q_{\boldsymbol{\phi}}(\mathbf{z}_0 \mid \mathbf{x})$ approaches $p_{\boldsymbol{\theta}}(\mathbf{Z}_T \mid \mathbf{x})$. As mentioned previously, we will learn an implicit distribution of $q_{\boldsymbol{\phi}}(\mathbf{z}_0 \mid \mathbf{x})$ (*i.e.*, learn how to draw samples from $q_{\boldsymbol{\phi}}(\mathbf{z}_0 \mid \mathbf{x})$ instead of its explicit form), as it allows us to chose a candidate distribution from a much larger distribution space, compared to explicitly defining $q_{\boldsymbol{\phi}}$[¶]. Consequently, $q_{\boldsymbol{\phi}}(\mathbf{z}_0 \mid \mathbf{x})$ is implemented by a stochastic generator (a DNN parameterized by $\boldsymbol{\phi}$) $Q_{\boldsymbol{\phi}}(\mathbf{z}_0 \mid \mathbf{x}, \omega)$ with input as the concatenation of $\mathbf{x}$ and $\omega$, where $\omega$ is a sample from an isotropic Gaussian distribution $q_0(\omega)$. Our goal is now translated to update the parameter $\boldsymbol{\phi}$ of $Q_{\boldsymbol{\phi}}(\mathbf{z}_0 \mid \mathbf{x}, \omega)$ to $\boldsymbol{\phi}'$ such that the distribution of $\{\mathbf{z}_0' = Q_{\boldsymbol{\phi}'}(\mathbf{z}_0' \mid \mathbf{x}, \omega)\}$ with $\omega \sim q_0(\omega)$ matches that of $\mathbf{z}_1$ in the original generating process with $\boldsymbol{\phi}$ in Figure 1. In this way, the generating process of $\mathbf{z}_1$ via $\mathcal{T}_1$ is *distilled* into the parameterized generator $Q_{\boldsymbol{\phi}}(\cdot)$, eliminating the need to do a specific transformation via $\mathcal{T}_1$ in testing, and thus is very efficient. Specifically, we update $\boldsymbol{\phi}'$ such that

$$\boldsymbol{\phi}' = \arg\min_{\boldsymbol{\phi}} \mathcal{D}\left(\{\mathbf{z}_0'^{(i)}\}, \{\mathbf{z}_1^{(i)}\}\right) , \tag{8}$$

where $\{\mathbf{z}_0'^{(i)}\}_{i=1}^S$ are a set of samples generated from $q_{\boldsymbol{\phi}'}(\mathbf{z}_0' \mid \mathbf{x})$ via $Q_{\boldsymbol{\phi}}(\cdot)$, and $\{\mathbf{z}_1^{(i)}\}_{i=1}^S$ are samples drawn by $\omega^i \sim q_0(\omega), \tilde{\mathbf{z}}_0^i = Q_{\boldsymbol{\phi}}(\cdot \mid \mathbf{x}, \omega^i), \mathbf{z}_1^{(i)} \sim \mathcal{T}_1(\tilde{\mathbf{z}}_0^i); \mathcal{D}(\cdot, \cdot)$ is a metric between samples such as the simple Euclidean distance or the more advanced Wasserstein distance (Arjovsky et al., 2017). The optimization is done by applying standard stochastic gradient descent (SGD). We call this procedure distilling knowledge from $\mathcal{T}_1$ to $Q_{\boldsymbol{\phi}}(\cdot)$.

---

[¶]This is distinct from our density-estimation framework described in the next section, where an explicit form is assumed at the beginning for practical needs.

After distilling knowledge from $\mathcal{T}_1$, we apply the same procedure for other transformations $\mathcal{T}_k (k > 1)$ sequentially. The final inference network, represented by $q_{\boldsymbol{\phi}}(\cdot \mid \mathbf{x})$, can then well approximate the continuous-time flows, e.g., the distribution of $\mathbf{z}_0 \sim q_{\boldsymbol{\phi}}(\cdot \mid \mathbf{x})$ is close to $\rho_T$ from the CTF. This concept is illustrated in Figure 2. According to Theorem 2, the number of updates for $\boldsymbol{\phi}$ in training is still bounded by $O(\epsilon^{-3/2})$ for an $\epsilon$-accurate MSE, however, inference in testing is significantly boosted since we do not need to simulate a long-time transformations as shown in Figure 1 (right).

**Updating $\boldsymbol{\theta}$**   Given $\boldsymbol{\phi}$, $\boldsymbol{\theta}$ can be updated by simply optimizing the ELBO in (7), where $\rho_T$ is approximated by $\bar{\rho}_T$ from the discretized CTF. Specifically, the expectation w.r.t. $\rho_T$ in (7) is approximated by a sample average from:

$$\mathbf{z}_0 \sim q_{\boldsymbol{\phi}}(\mathbf{z}_0 \mid \mathbf{x}), \mathbf{z}_1 \sim \mathcal{T}_1(\mathbf{z}_0), \mathbf{z}_2 \sim \mathcal{T}_2(\mathbf{z}_1), \cdots, \mathbf{z}_K \sim \mathcal{T}_K(\mathbf{z}_{K-1}) .$$

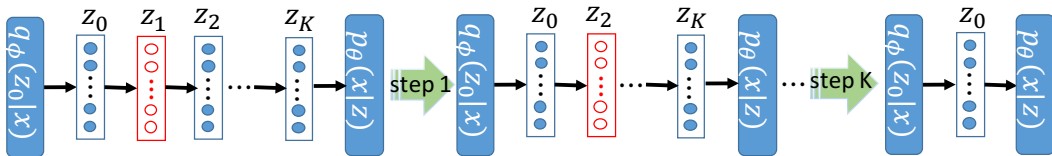

Figure 2: Amortized learning of continuous-time flows for VAEs. From left to right: the initial architecture with $K$-step transformations; For each step $k$, $q_{\boldsymbol{\phi}}(\cdot)$ is trained to match the distribuin of $\mathbf{z}_k$ in CTFs; In the end, the CTF is distilled into $q_{\boldsymbol{\phi}}(\cdot)$.

To sum up, there are three main steps in learning a CTF-based VAE:

1. Generate a sample path $(\mathbf{z}_0, \cdots, \mathbf{z}_K)$ according to $q_{\boldsymbol{\phi}}(\mathbf{z}_0 \mid \mathbf{x})$ and the discretized flow with transformations $\{\mathcal{T}_k\}$;
2. Update $\boldsymbol{\phi}$ according to (8);
3. Optimize $\boldsymbol{\theta}$ by minimizing the ELBO (7) with the generated sample path.

In testing, we use only the finally learned $q_{\boldsymbol{\phi}}(\mathbf{z}_0 \mid \mathbf{x})$ for inference (into which the CTF has been distilled), and hence testing is like the standard VAE. Since the discretized-CTF model is essentially a Markov chain, we call our model Markov-chain-based VAE (MacVAE).

## 4   CONTINUOUS TIME FLOWS FOR EXPLICIT DENSITY ESTIMATION

We describe how to apply the proposed CTF framework to density estimation of the observed data. We assume that the density of the observation $\mathbf{x}$ is characterized by a parametric Gibbsian-style probability model $p_{\boldsymbol{\theta}}(\mathbf{x}) = \frac{1}{\mathcal{Z}(\boldsymbol{\theta})}\tilde{p}_{\boldsymbol{\theta}}(\mathbf{x}) \triangleq \frac{1}{\mathcal{Z}(\boldsymbol{\theta})}e^{U(\mathbf{x};\boldsymbol{\theta})}$, where $\tilde{p}_{\boldsymbol{\theta}}(\mathbf{x})$ is an unnormalized version of $p_{\boldsymbol{\theta}}(\mathbf{x})$ with parameter $\boldsymbol{\theta}$, $U(\mathbf{x};\boldsymbol{\theta}) \triangleq \log \tilde{p}_{\boldsymbol{\theta}}(\mathbf{x})$ is called the energy function (Zhao et al., 2017), and $\mathcal{Z}(\boldsymbol{\theta}) \triangleq \int \tilde{p}_{\boldsymbol{\theta}}(\mathbf{x})\mathrm{d}\,\mathbf{x}$ is the normalizer. Note this form of distributions constitutes a very large class of distributions as long as the capacity of the energy function is large enough. This can be easily achieved by adopting a DNN to implement $U(\mathbf{x};\boldsymbol{\theta})$, the setting we considered in this paper. Note our model can be placed in between existing implicit and explicit density estimation methods, because we model the data density with an explicit distribution form up to an intractable normalizer. Such distributions have been proved to be useful in real applications, e.g., Haarnoja et al. (2017) used them to model policies in deep reinforcement learning.

Our goal is to learn $\boldsymbol{\theta}$ given observations $\{\mathbf{x}_i\}_{i=1}^N$, which can be achieved via the standard maximum likelihood estimator (MLE):

$$\boldsymbol{\theta} = \arg\max_{\boldsymbol{\theta}} \sum_{i=1}^N \log p_{\boldsymbol{\theta}}(\mathbf{x}_i) \triangleq \arg\max_{\boldsymbol{\theta}} \mathcal{M}(\{\mathbf{x}_i\}; \boldsymbol{\theta})$$

This is usually optimized via SGD, with the following gradient formula:

$$\frac{\partial \mathcal{M}(\{\mathbf{x}_i\}; \boldsymbol{\theta})}{\partial \boldsymbol{\theta}} = \frac{1}{N}\sum_{i=1}^N \frac{\partial U(\mathbf{x}_i; \boldsymbol{\theta})}{\partial \boldsymbol{\theta}} - \mathbb{E}_{p_{\boldsymbol{\theta}}(\mathbf{x})}\left[\frac{\partial U(\mathbf{x};\boldsymbol{\theta})}{\partial \boldsymbol{\theta}}\right] \tag{9}$$

---

**Algorithm 1** CTFs for generative models at the $k$-th iteration. $\mathcal{D}(\cdot, \cdot)$ is the same as (8).

---

**Input:** parameters from last step $\boldsymbol{\theta}^{(k-1)}, \boldsymbol{\phi}^{(k-1)}$
**Output:** updated parameters $\boldsymbol{\theta}^{(k)}, \boldsymbol{\phi}^{(k)}$
1. Generate samples $\{\mathbf{x}_{1,s}\}_{s=1}^S$ via a discretized CTF: $\mathbf{x}_{0,s} \sim q_{\boldsymbol{\phi}^{(k-1)}}(\mathbf{x}_0), \mathbf{x}_{1,s} \sim \mathcal{T}_1(\mathbf{x}_{0,s})$;
2. Update the generator by minimizing ($\{\mathbf{x}'_{0,s}\}_{s=1}^S$ are generated with the updated parameter $\boldsymbol{\phi}^{(k)}$):

$$\boldsymbol{\phi}^{(k)} = \arg\min_{\boldsymbol{\phi}} \mathcal{D}\left(\{\mathbf{x}_{1,s}\}, \{\mathbf{x}'_{0,s}\}\right) \ .$$

3. Update the energy-based model $\boldsymbol{\theta}^k$ by maximum likelihood, with gradient as (9) except replacing $\mathbb{E}_{\mathbf{x}\sim p_{\boldsymbol{\theta}}(\mathbf{x})}$ with $\mathbb{E}_{\mathbf{x}\sim q_{\boldsymbol{\phi}}(\mathbf{x})}$;

---

The gradient formula requires an integration over the model distribution $p_{\boldsymbol{\theta}}(\mathbf{x})$, which can be approximated by Monte Carlo integration with samples. The sampling problem has been well studied for some particular energy-based distributions, for example, via contrastive divergence in restricted Boltzmann machines (Hinton, 2002). However, this does not fit into our setting directly. Here we adopt the idea of CTFs and propose to use a DNN guided by a CTF, which we call a *generator*, to generate approximate samples from the original model $p_{\boldsymbol{\theta}}(\mathbf{x})$. Specifically, we require that samples from the generator should well approximate the target $p_{\boldsymbol{\theta}}(\mathbf{x})$. This can be done by adopting the CTF idea above, *i.e.*, distilling knowledge of a CTF (which approaches $p_{\boldsymbol{\theta}}(\mathbf{x})$) to the generator. In testing, instead of generating samples from $p_{\boldsymbol{\theta}}(\mathbf{x})$ via MCMC (which is complicated and time consuming), we generate samples from the generator directly. Furthermore, when evaluating the likelihood for test data, the unknown constant $\mathcal{Z}(\boldsymbol{\theta})$ of $p_{\boldsymbol{\theta}}(\mathbf{x})$ can also be approximated by Monte Carlo integration with samples drawn from the generator.

On the right side of (9), the first term is a model fit to observed data, and the (negative) second term is a model fit to synthetic data drawn from $p_{\boldsymbol{\theta}(\mathbf{x})}$; this is similar to the critic/discriminator in GANs (Arjovsky et al., 2017), but derived directly from the MLE. More connections are discussed below.

## 4.1 LEARNING VIA AMORTIZATION

Our goal is to learn a generator whose generated samples match those from the original model $p_{\boldsymbol{\theta}}(\mathbf{x})$, by adopting the amortization idea with CTF in the *inference* section above. Similar to inference, the generator is learned implicitly. However, we also learn an explicit density model for the data by SGD, with samples from the implicit generator to estimate gradients in (9). Note that in this case, the CTF is performed directly on the data space, instead of on latent-variable space as in previous sections. Specifically, the sampling procedure from the generator plus a continuous-time-flow transformation are written as:

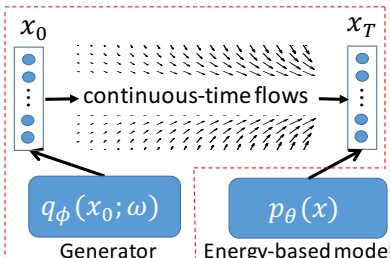

Figure 3: Learning a generator with CTF. The goal is to match the samples $\mathbf{x}_0$ from $q_{\boldsymbol{\phi}}$ to those after a CTF $(\mathbf{x}_T)$, or equivalently samples from $p_{\boldsymbol{\theta}}$.

$$\mathbf{x}_0 \sim q_{\boldsymbol{\phi}}(\mathbf{x}_0), \mathbf{x}_T \sim \mathcal{T}(\mathbf{x}_0, T) \ .$$

Here $\mathcal{T}(\cdot, \cdot)$ is the continuous-time flow; a sample $\mathbf{x}_0$ from $q_{\boldsymbol{\phi}}(\cdot)$ is implemented by a deep neural network (generator) $G_{\boldsymbol{\phi}}(\omega)$ with input $\omega \sim q_0(\omega)$, where $q_0$ is a simple distribution for a noise random variable, *e.g.*, the standard isotropic normal distribution. The procedure is illustrated in Figure 3. Note the CTF cannot be replaced by standard normalizing flow (Rezende & Mohamed, 2015) in this model, because there is no objective function to guide the update of parameters in normalizing flows, which is not necessary for CTFs.

Specifically, denote the parameters in the $k$-th step of our algorithm with subscript "$(k)$". For efficient sample generation, in the $k$-th step, we again adopt the amortization idea from Section 3.2 to update $\boldsymbol{\phi}^{(k-1)}$ of the generator network $G_{\boldsymbol{\phi}}(\cdot)$, such that samples from the updated generator match those from the current generator followed by a one-step transformation $\mathcal{T}_1(\cdot)$. After that, $\boldsymbol{\theta}$ is updated by drawing samples from $q_{\boldsymbol{\phi}}(\cdot)$ to estimate the expectation in (9). The detailed algorithm is presented in Algorithm 1.

### 4.2 Connections to Wasserstein GAN (WGAN) and MLE

There is an interesting relation between our model and the WGAN framework (Arjovsky et al., 2017). To see this, let $p_r$ be the data distribution. Substituting $p_{\boldsymbol{\theta}}(\mathbf{x})$ with $q_{\boldsymbol{\phi}}(\mathbf{x})$ for the expectation in the gradient formula (9) and integrating out $\boldsymbol{\theta}$, we have that our objective is

$$\max \mathbb{E}_{\mathbf{x} \sim p_r} [U(\mathbf{x}; \boldsymbol{\theta})] - \mathbb{E}_{\mathbf{x} \sim q_{\boldsymbol{\phi}}} [U(\mathbf{x}; \boldsymbol{\theta})] \tag{10}$$

The objective is an instance of the general integral probability metrics (Arjovsky & Bottou, 2017). When $U$ is chosen to be 1-Lipschitz functions, it recovers WGAN. This connection motivates us to introduce weight clipping (Arjovsky et al., 2017) or alternative regularizers (Gulrajani et al., 2017) when updating $\boldsymbol{\theta}$ for a better theoretical property. For this reason, we call our model Markov-chain-based GAN (MacGAN).

Furthermore, it can be shown by Jensen's inequality that the MLE is bounded by (detailed derivations are provided in Section C of the SM)

$$\max \frac{1}{N} \sum_{i=1}^{N} \log p_{\boldsymbol{\theta}}(\mathbf{x}_i) \leq \max \mathbb{E}_{\mathbf{x} \sim p_r} [U(\mathbf{x}; \boldsymbol{\theta})] - \mathbb{E}_{\mathbf{x} \sim q_{\boldsymbol{\phi}}} [U(\mathbf{x}; \boldsymbol{\theta})] - \mathbb{E}_{\mathbf{x} \sim q_{\boldsymbol{\phi}}} [\log q_{\boldsymbol{\phi}}] . \tag{11}$$

By inspecting (10) and (11), it is clear that: *i*) when learning the energy-based model parameters $\boldsymbol{\theta}$, the objective can be interpreted as maximizing an upper bound of the MLE shown in (11); *ii*) when optimizing the parameter $\boldsymbol{\phi}$ of the inference network, we adopt the amortized learning procedure presented in Algorithm 1, whose objective is $\min_{\boldsymbol{\phi}} \text{KL}(q_{\boldsymbol{\phi}} \| p_{\boldsymbol{\theta}})$, coinciding with the last two terms in (11). In other words, both $\boldsymbol{\theta}$ and $\boldsymbol{\phi}$ are optimized by maximizing the *same* upper bound of the MLE, guaranteeing convergence of the algorithm. Particularly, we can conclude that

**Proposition 4.** *The optimal solution of MacGAN is the maximum likelihood estimator.*

Note another difference between MacGAN and standard GAN framework is the way of learning the generator $q_{\boldsymbol{\phi}}$. We adopt the amortization idea, which directly guides $q_{\boldsymbol{\phi}}$ to approach $p_{\boldsymbol{\theta}}$; whereas in GAN, the generator is optimized via a min-max procedure to make it approach the empirical data distribution $p_r$. By explicitly learning $p_{\boldsymbol{\theta}}$, MacGAN is able to evaluate likelihood for test data (at least up to a constant).

## 5 Related Work

Our framework extends the idea of normalizing flows (Rezende & Mohamed, 2015) to continuous-time flows, by developing theoretical properties on the convergence behavior. Inference based on CTFs has been studied in Salimans et al. (2015) based on the auxiliary-variable technique. However, Salimans et al. (2015) directly uses discrete approximations for the flow, and the approximation accuracy is unclear. Moreover, the inference network requires simulating a long Markov chain for the auxiliary model, thus is less efficient than ours. Finally, the inference network is implemented as a parametric distribution (*e.g.*, the Gaussian distribution), limiting the representation power, a common setting in existing auxiliary-variable based models (Tran et al., 2016). The idea of amortization (Gershman & Goodman, 2014) has recently been explored in various research topics for Bayesian inference such as in variational inference (Kingma & Welling, 2014; Rezende et al., 2014) and Markov chain Monte Carlo (Wang & Liu, 2017; Li et al., 2017; Pu et al., 2017a). Both Wang & Liu (2017) and Pu et al. (2017a) extend the idea of Stein variational gradient descent (Liu & Wang, 2016) with amortized inference for a GAN-based and a VAE-based model, respectively, which resemble our proposed MacVAE and MacGAN in concept. Li et al. (2017) applies amortization to distill knowledge from MCMC to learn a student network. The ideas in Li et al. (2017) are similar to ours, but the motivation and underlying theory are different from that developed here.

## 6 Experiments

We conduct experiments to test our CTF-based framework for efficient inference and density estimation described above, and compared them with related methods. The implementation is based on the excellent code for SteinGAN$^{\|}$ Wang & Liu (2017), where we adopt their default parameter setting.

---

$^{\|}$https://github.com/DartML/SteinGAN

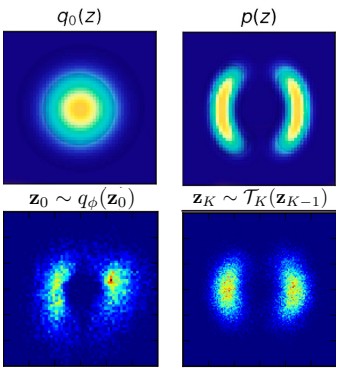
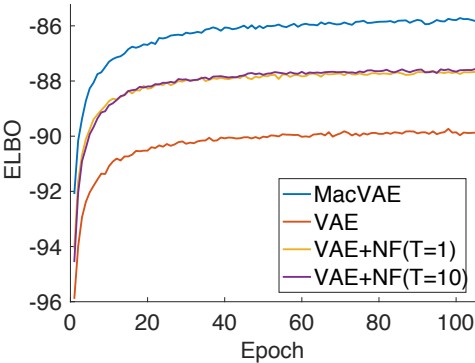

Figure 4: Knowledge distillation from the CTF (left) and ELBO versus epochs on MNIST (right). VAE with 80-layer NF is not included because it has much more parameters.

The discretization stepsize $h$ is robust as long as it is set in a reasonable range, *e.g.*, we set it the same as the stepsize in SGD.

## 6.1 CTFs FOR INFERENCE

**Synthetic experiment** We examine our amortized learning framework with a toy experiment. Following Rezende & Mohamed (2015), we use MacVAE to approximate samples from a two dimensional distribution on $\mathbf{z} = \{\mathbf{z}_1, \mathbf{z}_2\}$: $p(\mathbf{z}) \propto e^{-U(\mathbf{z})}$ with $U(\mathbf{z}) \triangleq \frac{1}{2}(\frac{\|\mathbf{z}\|-2}{0.4})^2 - \ln(e^{-\frac{1}{2}[\frac{\mathbf{z}_1-2}{0.6}]^2} + e^{-\frac{1}{2}[\frac{\mathbf{z}_1+2}{0.6}]^2})$. The inference network $q_{\boldsymbol{\phi}}$ is defined to be a 2-layer MLP with isotropic normal random variables as input. Figure 4 (top) plots the densities estimated with the samples from transformations $\{\mathcal{T}_{K=100}\}$ (before optimizing $\boldsymbol{\phi}$), as well as with samples generated directly from $q_{\boldsymbol{\phi}}$ (after optimizing $\boldsymbol{\phi}$). It is clear that the amortized learning is able to distill knowledge from the CTF to the inference network.

**MacVAE on MNIST** Following Rezende & Mohamed (2015); Tomczak & Welling (2016), we define the inference network as a deep neural network with two fully connected layers of size 300 with softplus activation functions. We compare MacVAE with the standard VAE and the VAE with normalizing flow, where testing ELBOs are reported (Section D.1 of the SM describes how to calculate the ELBO). We do not compare with other state-of-the-art methods such as the inverse autoregressive flow (Kingma et al., 2016), because they typically endowed more complicated inference networks (with more parameters), unfair for comparison. We use the same inference network architecture for all the models. Figure 4 (bottom) plots the testing ELBO versus training epochs. MacVAE outperforms VAE and normalizing flows with a better ELBO (around -85.62).

## 6.2 CTFs FOR DENSITY ESTIMATION

We test MacGAN on three datasets: MNIST, CIFAR-10 and CelabA. Following GAN-related methods, the model is evaluated by observing its ability to draw samples from the learned data distribution. Inspiring by Wang & Liu (2017), we define a parametric form of the energy-based model as $p_{\boldsymbol{\theta}}(\mathbf{x}) \propto \exp\{-\|\mathbf{x} - \text{DEC}_{\boldsymbol{\theta}}(\text{ENC}_{\boldsymbol{\theta}}(\mathbf{x}))\|^2\}$, where $\text{ENC}_{\boldsymbol{\theta}}(\cdot)$ and $\text{DEC}_{\boldsymbol{\theta}}(\cdot)$ are encoder and decoder defined by using deep convolutional neural networks and deconvolutional neural networks, respectively, parameterized by $\boldsymbol{\theta}$. For simplicity, we adopt the popular DCGAN architecture (Radford et al., 2016) for the encoder and decoder. The generator $G_{\boldsymbol{\phi}}$ is defined as a 3-layer convolutional neural network with the ReLU activation function (except for the top layer which uses tanh as the activation function, see SM D for details). Following Wang & Liu (2017), the stepsizes are set to $\frac{(m_e-e) \times l_r}{m_e-50}$, where $e$ indexes the epoch, $m_e$ is the total number of epochs, $l_r = $ 1e-4 when updating $\boldsymbol{\theta}$, and $l_r = $ 1e-3 when updating $\boldsymbol{\phi}$. The stepsize in $\mathcal{L}_1$ is set to 1e-3.

We compare MacGAN with DCGAN (Radford et al., 2016), the improved WGAN (WGAN-I) (Gulrajani et al., 2017) and SteinGAN (Wang & Liu, 2017). We plot images generated with MacGAN and its most related method SteinGAN in Figure 5 for CelebA and CIFAR-10 datasets. More results

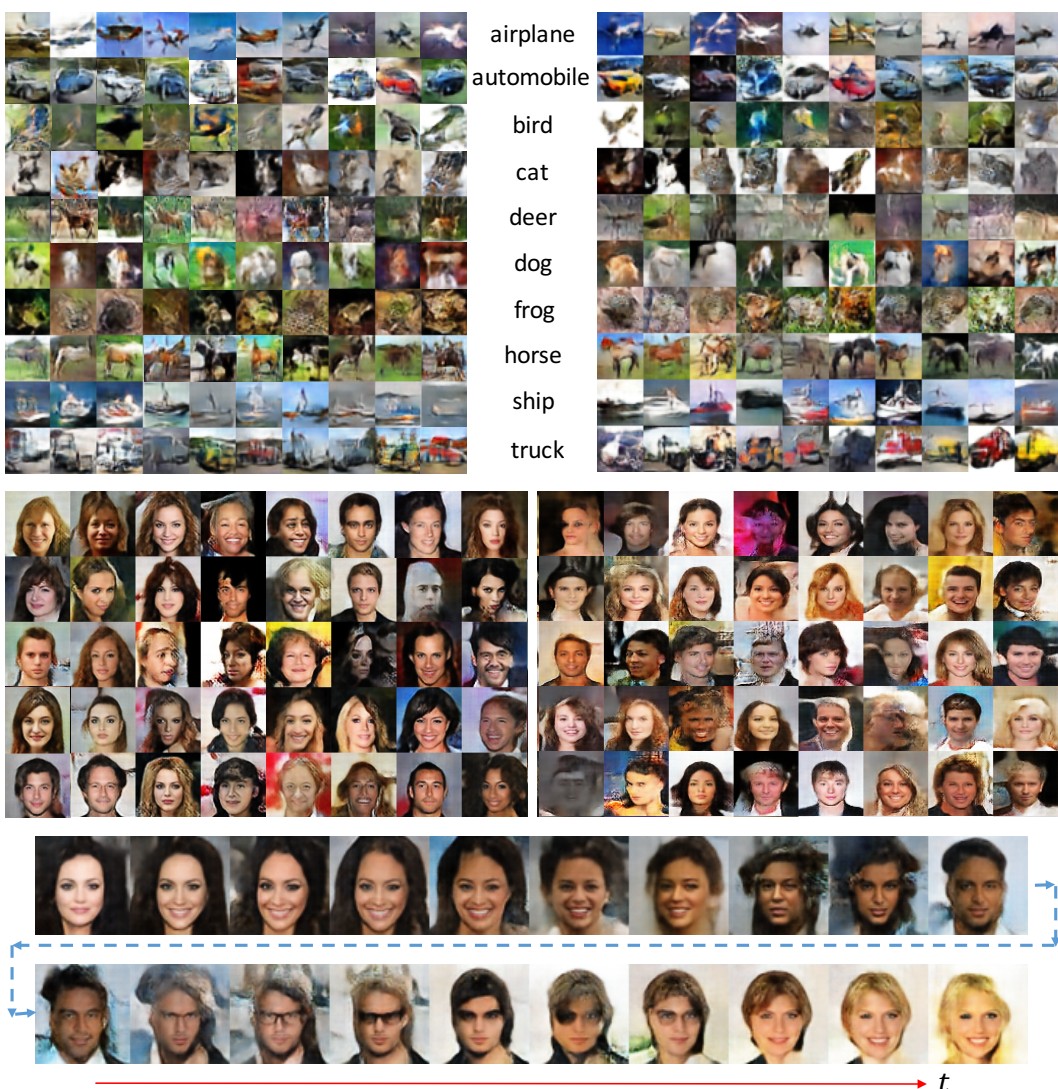

airplane
automobile
bird
cat
deer
dog
frog
horse
ship
truck

*t*

Figure 5: Generated images for CIFAR-10 (top) and CelebA (middle) datasets with MacGAN (left) and SteinGAN (right). The bottom are images generated by a random walk on the $\omega$ space for the generator of MacGAN, *i.e.*, $\omega_t = \omega_{t-1} + 0.03 \times \text{rand}([-1, 1])$.

are provided in SM Section D. We observe that visually MacGAN is able to generate clear-looking images. Following Wang & Liu (2017), we also plot the images generated by a random walk in the $\omega$ space in Figure 5.

Qualitatively evaluating a GAN-like model is challenging. We follow literature and use the inception score (Salimans et al., 2016) to measure the quantity of the generated images. Figure 6 plots inception scores versus training epochs for different models. MacGAN obtains competitive inception scores with the popular DCGAN model. Quantitatively, we get a final inception score of 6.49 for MacGAN, compared to 6.35 for SteinGAN, 6.25 for WGAN-I and 6.58 for DCGAN.

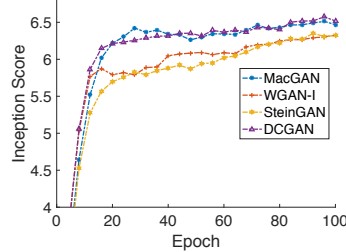

Figure 6: Inception score versus epochs for different models.

## 7 CONCLUSION

We study the problem of applying CTFs for efficient inference and explicit density estimation in deep generative models, two important tasks in unsupervised machine learning. Compared to discrete-time normalized flows, CTFs are more general and flexible due to the fact that their stationary distributions can be controlled without extra flow parameters. We develop theory on the approximation accuracy when adopting a CTF to approximate a target distribution. We apply CTFs on two classes of deep generative models, a variational autoencoder for efficient inference, and a GAN-like density estimator for explicit density estimation and efficient data generation. Experiments show encouraging results of our framework in both models compared to existing techniques. One interesting direction of future work is to explore more efficient learning algorithms for the proposed CTF-based framework.

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

# SUPPLEMENTARY MATERIAL FOR: CONTINUOUS-TIME FLOWS FOR EFFICIENT INFERENCE AND DENSITY ESTIMATION

## A ASSUMPTIONS OF THEOREM 2

First, let us define the infinitesimal generator of the diffusion (2). Formally, the *generator* $\mathcal{L}$ of the diffusion (2) is defined for any compactly supported twice differentiable function $f : \mathbf{R}^L \to \mathbf{R}$, such that,

$$\mathcal{L}f(\mathbf{Z}_t) \triangleq \lim_{h \to 0^+} \frac{\mathbb{E}\left[f(\mathbf{Z}_{t+h})\right] - f(\mathbf{Z}_t)}{h} = \left(F(\mathbf{Z}_t) \cdot \nabla + \frac{1}{2}\left(G(\mathbf{Z}_t)G(\mathbf{Z}_t)^T\right) : \nabla\nabla^T\right) f(\mathbf{Z}_t) ,$$

where $\mathbf{a} \cdot \mathbf{b} \triangleq \mathbf{a}^T \mathbf{b}$, $\mathbf{A} : \mathbf{B} \triangleq \mathrm{tr}(\mathbf{A}^T \mathbf{B})$, $h \to 0^+$ means $h$ approaches zero along the positive real axis.

Given an ergodic diffusion (2) with an invariant measure $\rho(\mathbf{Z})$, the posterior average is defined as: $\bar{\psi} \triangleq \int \psi(\mathbf{Z})\rho(\mathbf{Z})\mathrm{d}\,\mathbf{Z}$ for some test function $\psi(\mathbf{Z})$ of interest. For a given numerical method with generated samples $(\mathbf{z}_k)_{k=1}^K$, we use the *sample average* $\hat{\psi}$ defined as $\hat{\psi}_K = \frac{1}{K}\sum_{k=1}^K \psi(\mathbf{z}_k)$ to approximate $\bar{\psi}$. We define a functional $\tilde{\psi}$ that solves the following *Poisson Equation*:

$$\mathcal{L}\tilde{\psi}(\mathbf{z}_k) = \psi(\mathbf{z}_k) - \bar{\psi} \tag{12}$$

We make the following assumptions on $\tilde{\psi}$.

**Assumption 1.** *$\tilde{\psi}$ exists, and its up to 4rd-order derivatives, $\mathcal{D}^k\tilde{\psi}$, are bounded by a function $\mathcal{V}$, i.e., $\|\mathcal{D}^k\tilde{\psi}\| \leq C_k\mathcal{V}^{p_k}$ for $k = (0,1,2,3,4)$, $C_k, p_k > 0$. Furthermore, the expectation of $\mathcal{V}$ on $\{\mathbf{z}_k\}$ is bounded: $\sup_l \mathbb{E}\mathcal{V}^p(\mathbf{z}_k) < \infty$, and $\mathcal{V}$ is smooth such that $\sup_{s \in (0,1)} \mathcal{V}^p\left(s\,\mathbf{z} + (1-s)\,\mathbf{y}\right) \leq C\left(\mathcal{V}^p(\mathbf{z}) + \mathcal{V}^p(\mathbf{y})\right), \forall\,\mathbf{z},\mathbf{y}, p \leq \max\{2p_k\}$ for some $C > 0$.*

## B PROOFS FOR SECTION 3

*Sketch Proof of Lemma 1.* First note that (5) in Lemma 1 corresponds to eq.13 in Jordan et al. (1998), where $F(p)$ in Jordan et al. (1998) is in the form of $\mathrm{KL}(\rho\|p_{\boldsymbol{\theta}}(\mathbf{x},\mathbf{z}))$ in our setting.

Proposition 4.1 in Jordan et al. (1998) then proves that (5) has a unique solution. Theorem 5.1 in Jordan et al. (1998) then guarantees that the solution of (5) approach the solution of the Fokker-Planck equation in (3), which is $\rho_T$ in the limit of $h \to 0$.

Since this is true for each $k$ (thus each $t$ in $\rho_t$), we conclude that $\tilde{\rho}_k = \rho_{hk}$ in the limit of $h \to 0$. □

To prove Theorem 2, we first need a convergence result about convergence to equilibrium in Wasserstein distance for Fokker-Planck equations, which is presented in Bolley et al. (2012). Putting in our setting, we can get the following lemma based on Corollary 2.4 in Bolley et al. (2012).

**Lemma 5** (Bolley et al. (2012))**.** *Let $\rho_T$ be the solution of the FP equation (3) at time $T$, $p_{\boldsymbol{\theta}}(\mathbf{x},\mathbf{z})$ be the joint posterior distribution given $\mathbf{x}$. Assume that $\int \rho_T(\mathbf{z})p_{\boldsymbol{\theta}}^{-1}(\mathbf{x},\mathbf{z})\mathrm{d}\,\mathbf{z} < \infty$ and there exists a constant $C$ such that $\frac{\mathrm{d}W_2^2(\rho_T,p_{\boldsymbol{\theta}}(\mathbf{x},\mathbf{z}))}{\mathrm{d}t} \geq CW_2^2(\rho_T, p_{\boldsymbol{\theta}}(\mathbf{x},\mathbf{z}))$. Then*

$$W_2(\rho_T, p(\mathbf{x},\mathbf{z})) \leq W_2(\rho_0, p(\mathbf{x},\mathbf{z}))\, e^{-CT} . \tag{13}$$

We further need to borrow convergence results from Mattingly et al. (2010); Vollmer et al. (2016); Chen et al. (2015) to characterize error bounds of a numerical integrator for the diffusion (2). Specifically, the goal is to evaluate the posterior average of a test function $\psi(\mathbf{z})$, defined as $\bar{\psi} \triangleq$

$\int \psi(\mathbf{z})p_{\boldsymbol{\theta}}(\mathbf{x}, \mathbf{z})\mathrm{d}\,\mathbf{z}$. When using a numerical integrator to solve (2) to get samples $\{\mathbf{z}_k\}_{k=1}^K$, the sample average $\hat{\psi}_K \triangleq \frac{1}{K}\sum_{k=1}^K \psi(\mathbf{z}_k)$ is used to approximate the posterior average. The accuracy is characterized by the mean square error (MSE) defined as: $\mathbb{E}\left(\hat{\psi}_K - \bar{\psi}\right)^2$. Lemma 6 derives the bound for the MSE.

**Lemma 6** (Vollmer et al. (2016)). *Under Assumption 1, and for a 1st-order numerical intergrator, the MSE is bounded, for a constant $C$ independent of $h$ and $K$, by*

$$\mathbb{E}\left(\hat{\psi}_K - \bar{\psi}\right)^2 \leq C\left(\frac{1}{hK} + h^2\right) .$$

Furthermore, except for the 2nd-order Wasserstein distance defined in Lemma 1, we define the 1st-order Wasserstein distance between two probability measures $\mu_1$ and $\mu_2$ as

$$W_1\left(\mu_1, \mu_2\right) \triangleq \inf_{p \in \mathcal{P}(\mu_1, \mu_2)} \int \|\mathbf{x} - \mathbf{y}\|_2 \, p(\mathrm{d}\,\mathbf{x}, \mathrm{d}\,\mathbf{y}) . \tag{14}$$

According to the Kantorovich-Rubinstein duality Arjovsky et al. (2017), $W_1(\mu_1, \mu_2)$ is equivalently represented as

$$W_1\left(\mu_1, \mu_2\right) = \sup_{f \in \mathcal{L}_1} \mathbb{E}_{\mathbf{z} \sim \mu_1}\left[f(\mathbf{z})\right] - \mathbb{E}_{\mathbf{z} \sim \mu_2}\left[f(\mathbf{z})\right] , \tag{15}$$

where $\mathcal{L}_1$ is the space of 1-Lipschitz functions $f : \mathbb{R}^L \to \mathbb{R}$.

We have the following relation between $W_1(\mu_1, \mu_2)$ and $W_2(\mu_1, \mu_2)$.

**Lemma 7** (Givens & Shortt (1984)). *We have for any two distributions $\mu_1$ and $\mu_2$ that $W_1(\mu_1, \mu_2) \leq W_2(\mu_1, \mu_2)$.*

Now it is ready to prove Theorem 2.

*Proof of Theorem 2.* The idea is to simply decompose the MSE into two parts, with one part charactering the MSE of the numerical method, the other part charactering the MSE of $\rho_T$ and $p_{\boldsymbol{\theta}}(\mathbf{x}, \mathbf{z})$, which consequentially can be bounded using Lemma 5 above.

Specifically, we have

$$\begin{aligned}
\mathrm{MSE}(\bar{\rho}_T, \rho_T; \psi) &\triangleq \mathbb{E}\left(\int \psi(\mathbf{z})(\tilde{\rho}_T - \rho_T)(\mathbf{z})\mathrm{d}\,\mathbf{z}\right)^2 \\
&= \mathbb{E}\left(\frac{1}{K}\sum_{k=1}^K \psi(\mathbf{z}_k) - \int \psi(\mathbf{z})\rho_T(\mathbf{z})\mathrm{d}\,\mathbf{z}\right)^2 \\
&= \mathbb{E}\left(\left(\frac{1}{K}\sum_{k=1}^K \psi(\mathbf{z}_k) - \int \psi(\mathbf{z})p_{\boldsymbol{\theta}}(\mathbf{x}, \mathbf{z})\mathrm{d}\,\mathbf{z}\right) - \left(\int \psi(\mathbf{z})\rho_T(\mathbf{z})\mathrm{d}\,\mathbf{z} - \int \psi(\mathbf{z})p_{\boldsymbol{\theta}}(\mathbf{x}, \mathbf{z})\mathrm{d}\,\mathbf{z}\right)\right)^2 \\
&\overset{(1)}{=} \mathbb{E}\left(\frac{1}{K}\sum_{k=1}^K \psi(\mathbf{z}_k) - \int \psi(\mathbf{z})p_{\boldsymbol{\theta}}(\mathbf{x}, \mathbf{z})\mathrm{d}\,\mathbf{z}\right)^2 + \left(\int \psi(\mathbf{z})\rho_T(\mathbf{z})\mathrm{d}\,\mathbf{z} - \int \psi(\mathbf{z})p_{\boldsymbol{\theta}}(\mathbf{x}, \mathbf{z})\mathrm{d}\,\mathbf{z}\right)^2 \\
&\overset{(2)}{\leq} \mathbb{E}\left(\frac{1}{K}\sum_{k=1}^K \psi(\mathbf{z}_k) - \int \psi(\mathbf{z})p_{\boldsymbol{\theta}}(\mathbf{x}, \mathbf{z})\mathrm{d}\,\mathbf{z}\right)^2 + W_1^2(\rho_T, p_{\boldsymbol{\theta}}) \\
&\overset{(3)}{\leq} \mathbb{E}\left(\frac{1}{K}\sum_{k=1}^K \psi(\mathbf{z}_k) - \int \psi(\mathbf{z})p_{\boldsymbol{\theta}}(\mathbf{x}, \mathbf{z})\mathrm{d}\,\mathbf{z}\right)^2 + W_2^2(\rho_T, p_{\boldsymbol{\theta}}) \\
&\overset{(4)}{\leq} C_1\left(\frac{1}{hK} + h^2\right) + W_2^2\left(\rho_0, p(\mathbf{x}, \mathbf{z})\right)e^{-2CT} \\
&= O\left(\frac{1}{hK} + h^2 + e^{-2ChK}\right) ,
\end{aligned}$$

where "(1)" follows by the fact that $\mathbb{E}\left(\frac{1}{K}\sum_{k=1}^{K}\psi(\mathbf{z}_k) - \int \psi(\mathbf{z})p_{\boldsymbol{\theta}}(\mathbf{x}, \mathbf{z})\mathrm{d}\,\mathbf{z}\right) = 0$ Chen et al. (2015); "(2)" follows by the definition of $W_1(\mu_1, \mu_2)$ in (14) and the 1-Lipschitz assumption of the test function $\psi$; "(3)" follows by Lemma 7; "(4)" follows by Lemma 5 and Lemma 6. $\qquad\square$

## C  CONNECTION TO WGAN

We derive the upper bound of the maximum likelihood estimator, which connects MacGAN to WGAN. Let $p_r$ be the data distribution, rewrite our maximum likelihood objective as

$$\max \frac{1}{N}\sum_{i=1}^{N}\log p_{\boldsymbol{\theta}}(\mathbf{x}_i) = \max \frac{1}{N}\sum_{i=1}^{N}\left(U(\mathbf{x}_i; \boldsymbol{\theta}) - \log \int e^{U(\mathbf{x};\boldsymbol{\theta})}\mathrm{d}\,\mathbf{x}\right) .$$

The above maximum likelihood estimator can be bounded with Jensen's inequality as:

$$\max \frac{1}{N}\sum_{i=1}^{N}\log p_{\boldsymbol{\theta}}(\mathbf{x}_i) \leq \max \mathbb{E}_{\mathbf{x}\sim p_r}\left[U(\mathbf{x}; \boldsymbol{\theta})\right] - \log \int \frac{e^{U(\mathbf{x};\boldsymbol{\theta})}}{q_{\boldsymbol{\phi}}(\mathbf{x};\omega)}q_{\boldsymbol{\phi}}(\mathbf{x};\omega)\mathrm{d}\,\mathbf{x}$$

$$\leq \max \mathbb{E}_{\mathbf{x}\sim p_r}\left[U(\mathbf{x}; \boldsymbol{\theta})\right] - \mathbb{E}_{\mathbf{x}\sim q_{\boldsymbol{\phi}}(\mathbf{x};\omega)}\left[\log \frac{e^{U(\mathbf{x};\boldsymbol{\theta})}}{q_{\boldsymbol{\phi}}(\mathbf{x};\omega)}\right]$$

$$= \max \mathbb{E}_{\mathbf{x}\sim p_r}\left[U(\mathbf{x}; \boldsymbol{\theta})\right] - \mathbb{E}_{\mathbf{x}\sim q_{\boldsymbol{\phi}}(\mathbf{x};\omega)}\left[U(\mathbf{x}; \boldsymbol{\theta})\right] - \mathbb{E}_{\mathbf{x}\sim q_{\boldsymbol{\phi}}(\mathbf{x};\omega)}\left[\log q_{\boldsymbol{\phi}}(\mathbf{x};\omega)\right] . \qquad (16)$$

This results in the same objective form as WGAN except that our model does not restrict $U(\mathbf{x}; \boldsymbol{\theta})$ to be 1-Lipschitz functions and the objective has an extra constant term $\mathbb{E}_{\mathbf{x}\sim q_{\boldsymbol{\phi}}(\mathbf{x};\omega)}\left[\log q_{\boldsymbol{\phi}}(\mathbf{x};\omega)\right]$ w.r.t. $\boldsymbol{\theta}$.

Now we prove Proposition 4.

*Proof of Proposition 4.* First it is clear that the equality in (16) is achieved if and only if

$$q_{\boldsymbol{\phi}}(\mathbf{x};\omega) = p_{\boldsymbol{\theta}}(\mathbf{x}) \propto e^{U(\mathbf{x};\boldsymbol{\theta})} .$$

From the description in Section 4 and (16), we know that $\boldsymbol{\theta}$ and $\boldsymbol{\phi}$ share the same objective function, which is an upper bound of the MLE in (16).

Furthermore, based on the property of continuous-time flows (or formally Theorem 2), we know that $q_{\boldsymbol{\phi}}$ is learned such that $q_{\boldsymbol{\phi}} \to p_{\boldsymbol{\theta}}$ in the limit of $h \to 0$ (or alternatively, we could achieve this by using a decreasing-step-size sequence in a numerical method, as proved in Chen et al. (2015)). When $q_{\boldsymbol{\phi}} = p_{\boldsymbol{\theta}}$, the equality in (16) is achieved, leading to the MLE. $\qquad\square$

## D  ADDITIONAL EXPERIMENTS

### D.1  CALCULATING THE TESTING ELBO FOR MACVAE

We follow the method in Pu et al. (2017a) for calculating the ELBO for a test data $\mathbf{x}_*$. First, after distilling the CTF into the inference network $q_{\boldsymbol{\phi}}$, we have that the ELBO can be represented as

$$\log p(\mathbf{x}_*) \geq \mathbb{E}_{q_{\boldsymbol{\phi}}}\left[\log p_{\boldsymbol{\theta}}(\mathbf{x}_*, \mathbf{z}_*)\right] - \mathbb{E}_{q_{\boldsymbol{\phi}}}\left[\log q_{\boldsymbol{\phi}}\right] .$$

The expectation is approximated with samples $\{\mathbf{z}_{*j}\}_{j=1}^{M}$ with $\mathbf{z}_{*j} = f_{\boldsymbol{\phi}}(\mathbf{x}_*, \boldsymbol{\zeta}_j)$, and $\boldsymbol{\zeta}_j \sim q_0(\boldsymbol{\zeta})$ the standard isotropic normal. Here $f_{\boldsymbol{\phi}}$ represents the deep neural network in the inference network. Note $q_{\boldsymbol{\phi}}(\mathbf{z}_*)$ is not readily obtained. To evaluate it, we use the density transformation formula: $q_{\boldsymbol{\phi}}(\mathbf{z}_*) = q_0(\boldsymbol{\zeta})\left|\det\frac{\partial f_{\boldsymbol{\phi}}(\mathbf{x}_*, \boldsymbol{\zeta})}{\partial \boldsymbol{\zeta}}\right|^{-1}$.

### D.2  NETWORK ARCHITECTURE

The architecture of the generator of MacGAN is given in Table 1.

Table 1: Architecture of generator in MacGAN

| Output Size | Architecture |
|---|---|
| $100 \times 1$ | $100 \times 10$ Linear, BN, ReLU |
| $256 \times 8 \times 8$ | $512 \times 4 \times 4$ deconv, $256$ $5 \times 5$ kernels, ReLU, strike 2, BN |
| $128 \times 16 \times 16$ | $256 \times 8 \times 8$ deconv, $128$ $5 \times 5$ kernels, ReLU, strike 2, BN |
| $3 \times 32 \times 32$ | $128 \times 16 \times 16$ deconv, $3$ $5 \times 5$ kernels, Tanh, strike 2 |

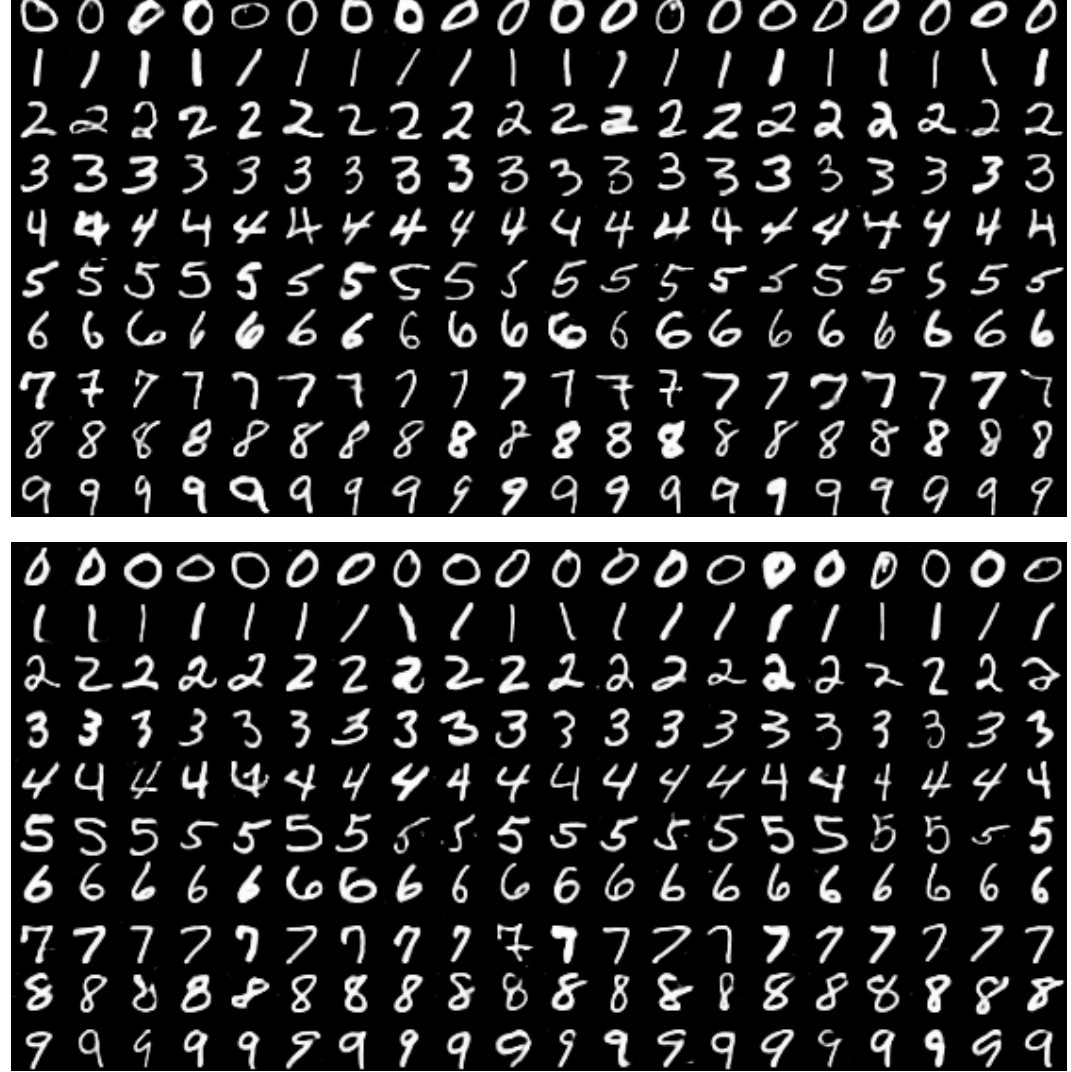

Figure 7: Generated images for MNIST datasets with MacGAN (top) and SteinGAN (bottom).

### D.3 ADDITIONAL RESULTS

Additional experimental results are given in Figure 7 – 12.

### D.4 ROBUSTNESS OF THE DISCRETIZATION STEPSIZE

To test the impact of the discretization stepsize $h$ in (6), following SteinGAN Feng et al. (2017), we test MacGAN on the MNIST dataset, where ee use a simple Gaussian-Bernoulli Restricted Boltzmann Machines as the energy-based model. We adopt the annealed importance sampling method to evaluate

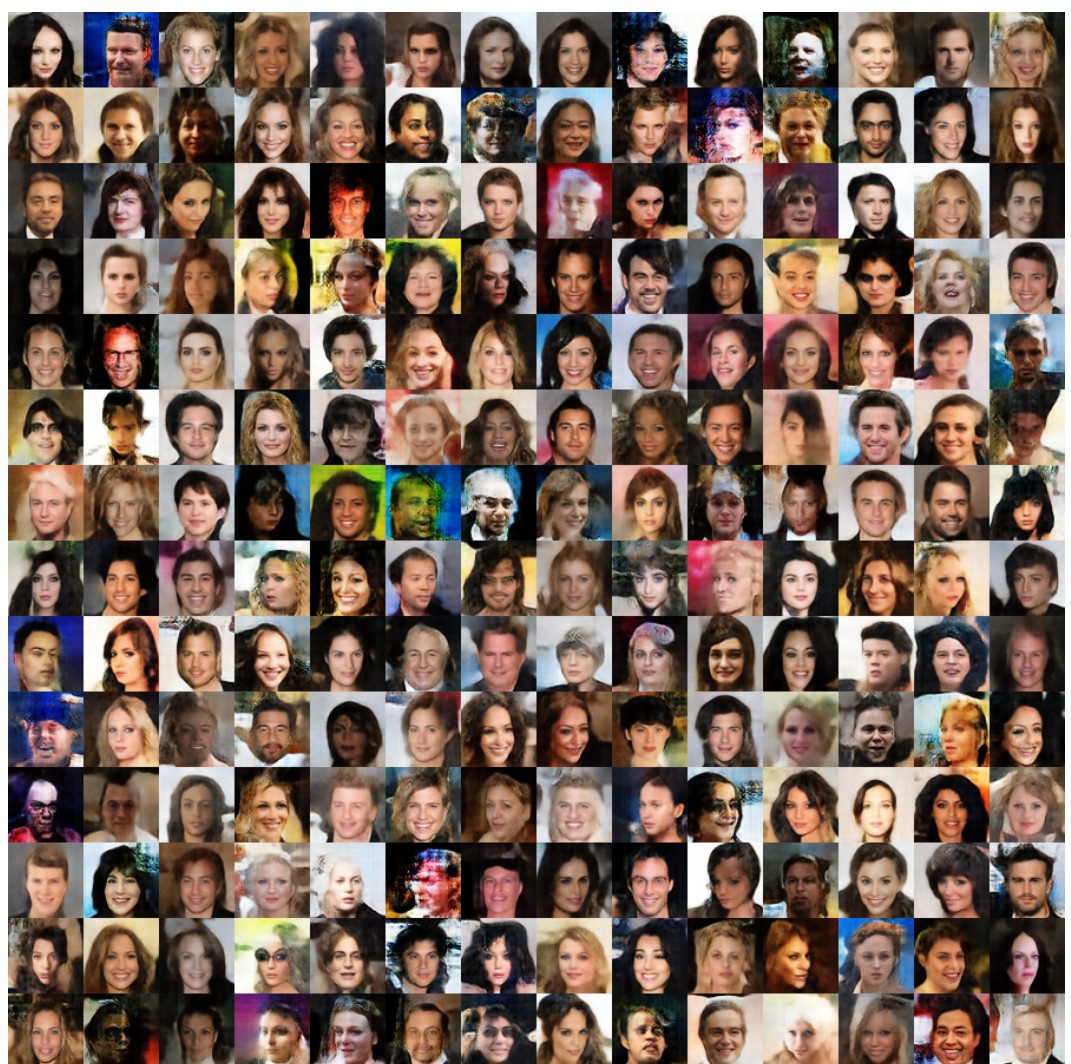

Figure 8: Generated images for CelebA datasets with MacGAN.

log-likelihoods Feng et al. (2017). We vary $h$ in $\{6e-4, 2.4e-3, 3.6e-3, 6e-3, 1e-2, 1.5e-2\}$. The trend of log-likelihoods is plotted in Figure 13. We can see that log-likelihoods do not change a lot within the chosen stepsize interval, demonstrating the robustness of $h$.

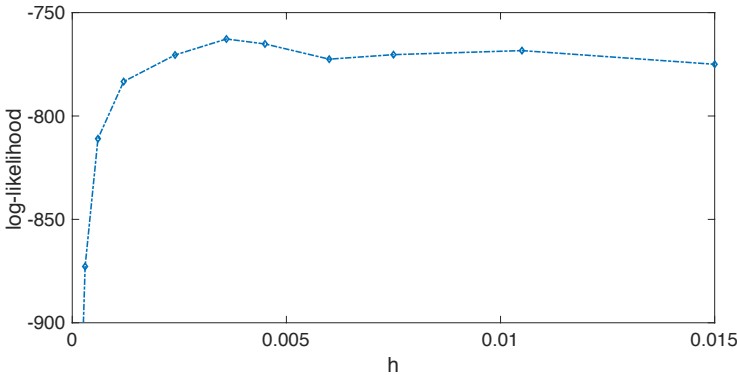

Figure 13: Log-likelihoods vs discretization stepsize for MacGAN on MNIST.

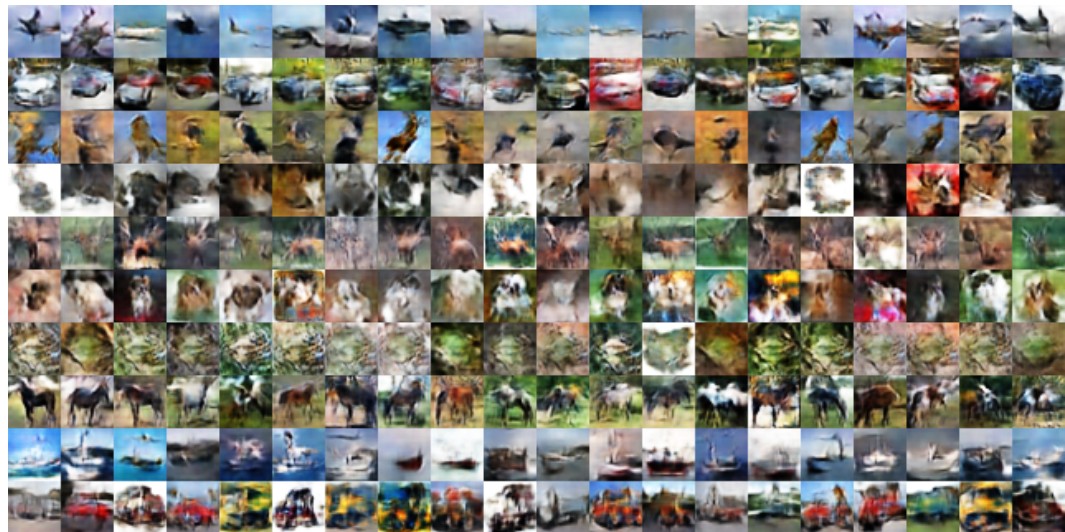

Figure 9: Generated images for CIFAR-10 datasets with MacGAN.

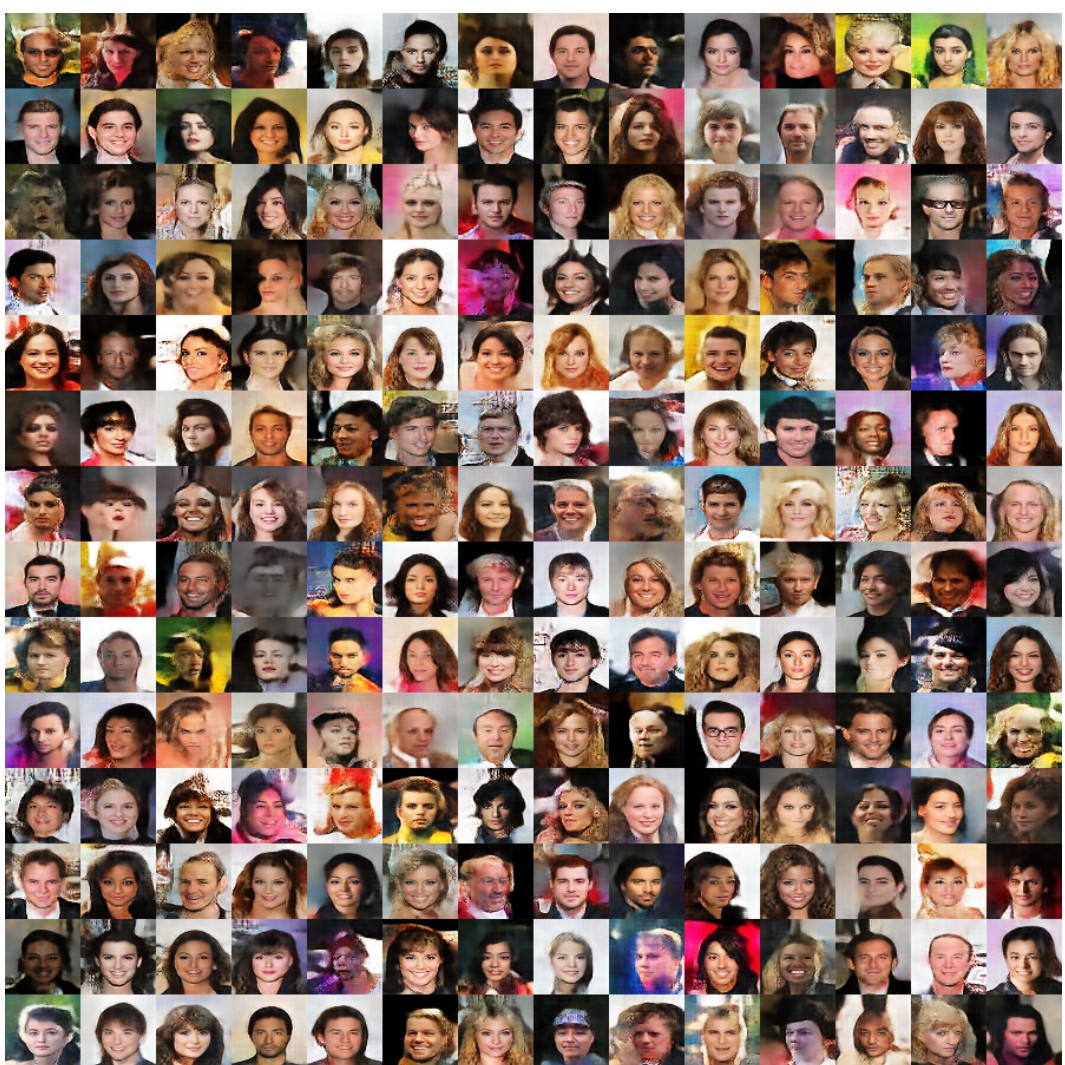

Figure 10: Generated images for CelebA datasets with SteinGAN.

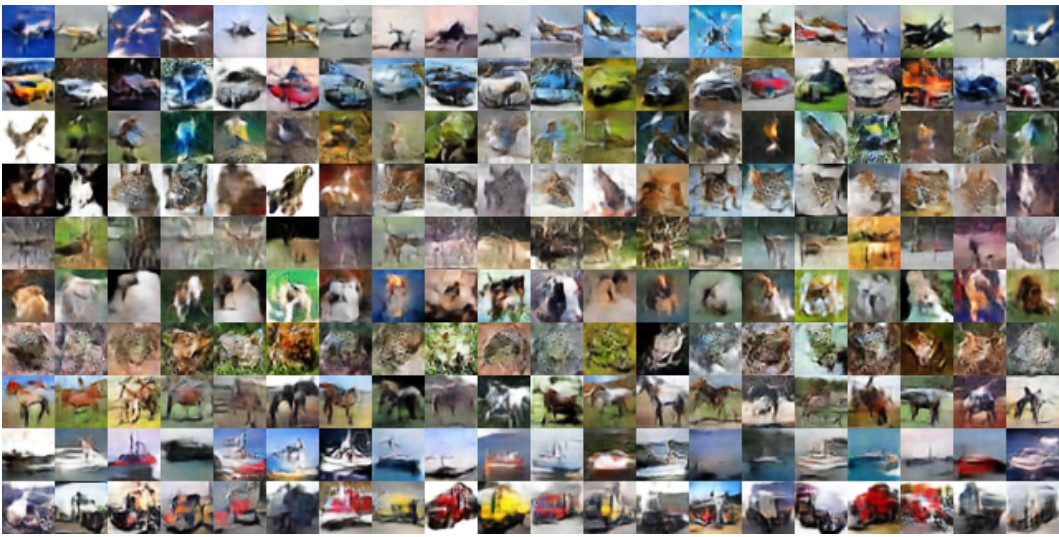

Figure 11: Generated images for CIFAR-10 datasets with SteinGAN.

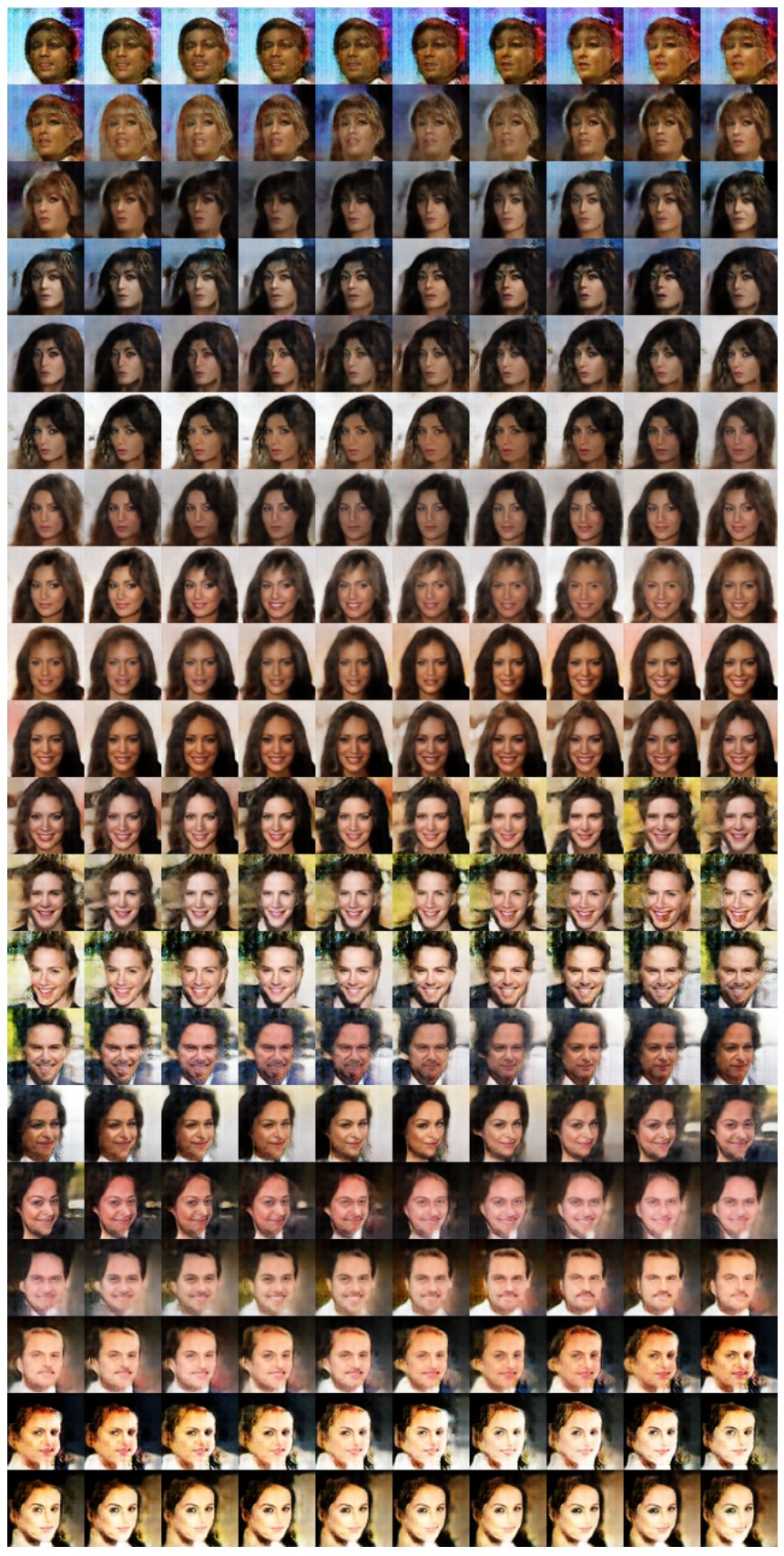

Figure 12: Generated images with a random walk on the $\omega$ space for CelebA datasets with MacGAN, $\omega_t = \omega_{t-1} + 0.02 \times \text{rand}([-1, 1])$.

