# OpenReview forum: "Continuous-Time Flows for Efficient Inference and Density Estimation"
_ICLR.cc/2018/Conference — Reject_

### Official Review · AnonReviewer1 · 2017-11-24
**Not convinced!**

**Rating:** 3
**Confidence:** 3

**Review:**

The authors try to use continuous time generalizations of normalizing flows for improving upon VAE-like models or for standard density estimation problems.

Clarity: the text is mathematically very sloppy / hand-wavy.

1. I do not understand proposition (1). I do not think that the proof is correct (e.g. the generator L needs to be applied to a function -- the notation L(x) does not make too much sense): indeed, in the case when the volatility is zero (or very small), this proposition would imply that any vector field induces a volume preserving transformation, which is indeed false.

2. I do not really see how the sequence of minimization Eq(5) helps in practice. The Wasserstein term is difficult to hand.

3. in Equation (6), I do not really understand what $\log(\bar{\rho})$ is if $\bar{\rho}$ is an empirical distribution. One really needs $\bar{\rho}$ to be a probability density to make sense of that.

---

> ### Author Response · Authors · 2017-12-21
> **Thank you for valuable feedback, we have updated our draft**
>
> Thank you for valuable feedback, which make us aware of some presentation issues of our original submission. We hope we could engage in constructive discussions to fully clarify and address your concerns and questions. We have fixed the problems by re-writing Section 3, and hopefully address your concerns. We wish to take the opportunities to emphasize that the main proposed methodologies/algorithms are still valid, and the pointed problems are only relates to the writing. Below are our initial responses to your three comments.
>
> 1. You are right. Proposition 1 is not correct for all CTFs, but it is correct for some specific CTFs such as the Hamiltonian flow. Sorry for the mistake, we have removed it and re-written this section. Note that the proposed algorithm for inference does not rely on this proposition. This is because the Jacobian term is only necessary in explicit methods ( i.e. maintaining distribution forms) in representing the normalizing flows, while our amortized approach is implicit (i.e. sample-based approximation) in representing flows at each step. Please see Section 3.2 on the detailed learning algorithm.
>
> 2. Eq.5 is not directly implemented in practice, We have clarified it in our revision to avoid confusion.  Eq.5 is derived from the principle theory of CTF, and presented in the paper to justify (1) some potential advantages of using CTF and (2) the sequential procedure of approximating the unknown \rho_T.
>
> In practice, we build the algorithm on the sequential procedure in Eq.5, and amortized the inference in an implicit manner. Specifically, at each step, we (1) first simulated samples from the corresponding diffusion, which is equivalent to optimizing one step in Eq.5, i.e., the resulting sample distribution (implicit) equals that from optimizing eq.5, and (2) proposed to use a neural network to match (i.e., “distill the knowledge") the simulated sample distributions. Directly handling the optimization problem to obtained its explicit distribution forms is an interesting direction of future work.
>
> 3. In the original Eq.6, we meant to show the ELBO, assuming \bar{\rho} is continuous (in the infinite-data setting). We agree this is a little misleading, thus we have removed it, and reformulated the objective in our revision (still Eq.6). Thanks for pointing out this issue.

---

### Official Review · AnonReviewer2 · 2017-11-27
**Previous reviewer; interesting ideas furthering continuous-time flows**

**Rating:** 6
**Confidence:** 4

**Review:**


The authors propose continuous-time flows as a flexible family of
distributions for posterior inference of latent variable models as
well as explicit density estimation. They build primarily on the work
of normalizing flows from Rezende and Mohamed (2015). They derive an
interesting objective based on a sequence of sub-optimization
problems, following a variational formulation of the Fokker-Planck
equations.

I reviewed this paper for NIPS with a favorable decision toward weak
acceptance; and the authors also addressed some of my questions in
this newer version (namely, some comparisons to related work; clearer
writing).

The experiments are only "encouraging"; they do not illustrate clear
improvements over previous methods. However, I think the work
demonstrates useful ideas furthering the idea of continuous-time
transformations that warrants acceptance.

---

> ### Author Response · Authors · 2017-12-24
> **Thanks for positive feeback**
>
> We appreciate your consistent support for our work. The draft is updated again, hopefully it could be easier for the future readers to understand.

---

### Official Review · AnonReviewer3 · 2017-11-27
**No great novelty, but above the threshold.**

**Rating:** 6
**Confidence:** 4

**Review:**

The authors propose the use of first order Langevin dynamics as a way to transition from one latent variable to the next in the VAE setting, as opposed to the deterministic transitions of normalizing flow. The extremely popular Fokker-Planck equation is used to analyze the steady state distributions in this setting. The authors also propose the use of CTF in density estimation, as a generator of samples from the ''true'' distribution, and show competitive performance w.r.t. inception score for some common datasets.

The use of Langevin diffusion for latent transitions is a good idea in my opinion; though quite simple, it has the benefit of being straightforward to analyze with existing machinery. Though the discretized Langevin transitions in \S 3.1 are known and widely used, I liked the motivation afforded by Lemma 2.

I am not convinced that taking \rho to be the sample distribution with equal probabilities at the z samples is a good choice in \S 3.1; it would be better to incorporate the proximity of the langevin chain to a stationary point in the atom weights instead of setting them to 1/K. However to their credit the authors do provide an estimate of the error in the distribution stemming from their choice.

To the best of my knowledge the use of CTF in density estimation as described in \S 4 is new, and should be of interest to the community; though again it is fairly straightforward. Regarding the experiments, the difference in ELBO between the macVAE and the vanilla ones with normalizing flows is only about 2%; I wish the authors included a discussion on how the parameters of the discretized Langevin chain affects this, if at all.

Overall I think the theory is properly described and has a couple of interesting formulations, in spite of being not particularly novel. I think CTFs like the one described here will see increased usage in the VAE setting, and thus the paper will be of interest to the community.

---

> ### Author Response · Authors · 2017-12-21
> **Thank you for your recognition**
>
> Thank you for recognizing our work. We are happy to address the two questions raised.
>
> We agree that our way to approximate \rho_T is not optimal. We use the simple sample averaging for the convenience of analysis. Better approximation by assigning more weights to the more recent samples leads to more challenges in theoretical analysis. We have added some discussion about this in the Section 3.1.
>
> The stepsize parameter of the discretized Langevin chain does not affect model performance a lot as long as the stepsize lies in an appropriate range. To verify this, following SteinGAN with a simple Gaussian-Bernoulli Restricted Boltzmann Machines as the energy-based model (https://github.com/DartML/SteinGAN), we conducted an extra experiment on the MNIST dataset with MacGAN. We used the annealed importance sampling to evaluate log-likelihoods. Below are the log-likelihoods by varying the stepsize. More details are included in the appendix D.4.
>
> stepsize: 		6e-4	2.4e-3	3.6e-3	6e-3	1e-2	1.5e-2
> log-likelihood:	-800	-760	-752	-762	-758	-775

---

### Decision · Program_Chairs · 2018-01-29
**ICLR 2018 Conference Acceptance Decision**

**Decision:**

Reject

**Comment:**

Thank you for submitting you paper to ICLR. The consensus from the reviewers is that there are some interesting theoretical contributions and some promising experimental support. However, although the paper is moving in the right direction, they believe that it is not quite ready for publication.